# CD8$^+$ T cell-mediated endotheliopathy is a targetable mechanism of neuro-inflammation in Susac syndrome

Catharina C. Gross ⬤ et al.#

Neuroinflammation is often associated with blood-brain-barrier dysfunction, which contributes to neurological tissue damage. Here, we reveal the pathophysiology of Susac syndrome (SuS), an enigmatic neuroinflammatory disease with central nervous system (CNS) endotheliopathy. By investigating immune cells from the blood, cerebrospinal fluid, and CNS of SuS patients, we demonstrate oligoclonal expansion of terminally differentiated activated cytotoxic CD8$^+$ T cells (CTLs). Neuropathological data derived from both SuS patients and a newly-developed transgenic mouse model recapitulating the disease indicate that CTLs adhere to CNS microvessels in distinct areas and polarize granzyme B, which most likely results in the observed endothelial cell injury and microhemorrhages. Blocking T-cell adhesion by anti-α4 integrin-intervention ameliorates the disease in the preclinical model. Similarly, disease severity decreases in four SuS patients treated with natalizumab along with other therapy. Our study identifies CD8$^+$ T-cell-mediated endotheliopathy as a key disease mechanism in SuS and highlights therapeutic opportunities.

#A full list of authors and their affiliations appears at the end of the paper.

Brain endothelial cells (ECs) are major elements of the blood–brain barrier (BBB), which contribute to central nervous system (CNS) homeostasis, maintenance, and neuronal function by restricting the entry of circulating leukocytes and blood-derived molecules into the CNS[1,2]. Compromised function of brain ECs resulting in impaired integrity of the BBB is an early hallmark of various neurological diseases. These include not only autoimmune inflammatory disorders, such as multiple sclerosis (MS), neuromyelitis optica, and Rasmussen encephalitis, but also infections such as cerebral malaria and arbovirus-related encephalitis, as well as cerebrovascular diseases, such as ischemic stroke[3]. Upon their active entry into the CNS, leukocytes can contribute to lesion development[4,5]. Although several mechanisms targeting brain ECs and resulting in BBB dysfunction have been described, the impact and function of lymphocytes during these processes are still poorly understood[2].

To gain further insights into the underlying mechanisms resulting in BBB dysfunction caused by lymphocytes, we investigate the pathophysiology of Susac syndrome (SuS)[6], which is considered an inflammatory endotheliopathy. SuS is a rare neurological disease mainly affecting young adults between 20 and 40 years, with a female-to-male ratio of 3.5/1[7]. Since its first description in 1979[8], approximately 400 cases have been reported worldwide[7]. SuS patients present with a clinical triad of encephalopathy, visual disturbances caused by branch retinal artery occlusions, and sensorineural hearing deficits[7]. Histopathology studies suggest that the clinical manifestations are caused by focal microangiopathy affecting the small-to-medium-size vessels of the brain, retina, and inner ear[9,10].

The pathogenesis of SuS remains enigmatic. An (auto)immune process leading to the disruption and occlusion of microvessels in the affected organs has been postulated[11]. Elevated serum levels of anti-EC antibodies (AECA) are found in approximately 25% of SuS patients[12,13], suggestive of a pathogenic scenario involving an antibody-mediated attack against ECs. However, this hypothesis has been challenged, since AECA are only detected in a subset of patients and are not associated with disease severity[12]. Furthermore, few CD20[+] B cells and no plasma cells are present in CNS biopsies of SuS patients[9]. However, the clear response of SuS to immunosuppressive and immunomodulatory drugs[13–15] strongly supports the hypothesis that SuS is immune mediated and scant perivascular T lymphocytes have been demonstrated histopathologically[9].

Here, by combining in-depth immune profiling and phenotyping of blood and cerebrospinal fluid (CSF) samples with a pathological study of brain tissue from patients with SuS[6] or its differential diagnosis MS, we reveal distinct underlying mechanisms in these two chronic neuroinflammatory diseases. Furthermore, we mimic the scenario of an antigen-specific CD8[+] T cell attack against CNS endothelium in a mouse model. Taken together, we demonstrate that cytotoxic CD8[+] T cells (CTLs) can cause vascular CNS injury and identify CTL-mediated endotheliopathy as a targetable mechanism in SuS.

## Results

### Intrathecal and circulating CD8[+] T cells are altered in SuS.
In accordance with previous case reports[9,10,16], immunohistochemical characterization of brain tissue specimen from seven SuS patients revealed the presence of immune cells (Fig. 1a). Quantification showed that the majority of CNS infiltrates in SuS were composed of CD3[+] T cells, mainly consisting of CD8[+] T cells ranging from 56% to 89% of T cells. Contrasting with the density of CD8[+] T cells (mean density 14 CD8[+] T cells mm$^{-2}$; ranging from 2.3 to 73.2 CD8[+] T cells mm$^{-2}$), few CD20[+] B cells were detected in brain parenchyma (mean density 0.4 CD20[+] B

cells mm$^{-2}$), and no antibody-producing CD138[+] plasma cells were observed.

We further assessed SuS-specific changes in the intrathecal compartment by multi-parameter flow cytometry CSF analysis (Fig. 1b, Supplementary Fig. 1a, Supplementary Table 1). Immune profiles of patients with SuS were compared to MS patients, one of the major differential diagnosis of SuS. Furthermore, patients with neurological manifestations who could not be ascribed to any medical condition (somatoform disorders) and in the absence of an inflammatory CSF served as a non-inflammatory control group. Although 36% of the SuS patients displayed a disruption of the blood–CSF barrier, CSF cell counts in SuS patients remained within normal range (2.4 ± 4.6 cells µl$^{-1}$ CSF, Supplementary Table 2). In contrast, MS patients are usually characterized by mild pleocytosis (6.8 ± 8.8 cells µl$^{-1}$ CSF) as previously reported[17–19], resulting in a significant increase of all lymphocyte subsets studied, including T and B cells (Supplementary Fig. 2a). Although the proportion of CD19[+] B cells was slightly increased in the CSF of SuS patients, no antibody-producing CD138[+] plasma cells were found; the latter being a typical feature of MS CSF (Fig. 1b). Accordingly, intrathecal immunoglobulin (Ig) synthesis and presence of oligoclonal bands were present in the majority of MS cases but absent in SuS (Supplementary Table 2). Notably, owing to both increased proportion of CD8[+] T cells and decreased proportion of CD4[+] T cells, the intrathecal CD4/CD8 ratio was decreased in SuS patients compared to both non-inflammatory controls and MS patients (Fig. 1b). The proportion of activated HLA-DR-expressing CD8[+] T cells was significantly increased both in the CSF and in the peripheral blood of SuS patients, a feature also not shared by MS patients. Moreover, absolute numbers of HLA-DR[+] CD8[+] T cells were also increased in the CSF of SuS patients (Supplementary Fig. 2a). Of note, no differences in the peripheral as well as intrathecal T cell compartment were detected between treatment-naive SuS patients and those on corticosteroids (Supplementary Fig. 2b). The frequency of circulating CD45RA[+]CD62L[−] terminally differentiated effector memory CD8[+] T cells (CD8[+] T$_{EMRA}$), but not CD4[+] T$_{EMRA}$ cells, was significantly increased in SuS patients (Fig. 1c, Supplementary Fig. 1b), suggesting that a strong and prolonged antigenic stimulus had triggered CD8[+] T cell differentiation[20]. This feature was characteristic for SuS and unlike healthy controls showed no positive correlation with age (Supplementary Fig. 3a). In SuS, a trend for greater proportion of CD8[+] T$_{EMRA}$ cells was detected close to disease onset, suggesting that these cells may contribute to disease development (Supplementary Fig. 3b).

### CD8[+] T cells in SuS contain disease-specific expanded clones.
To explore whether the observed activation and differentiation of CD8[+] T cells was antigen driven, the T cell receptor (TCR) repertoire was first analyzed by Vβ complementarity-determining region 3 (Vβ-CDR3) spectratyping[21] (Fig. 2a). Compared to healthy individuals, SuS patients exhibited a decreased complexity score of CD8[+] T cells as well as an increase in the oligoclonal profile of the Vβ-CDR3 spectratypes, indicating a significant clonal expansion in the CD8[+] T cell compartment. In contrast, despite a more skewed phenotype, no oligoclonal expansion was detectable with this technique in the peripheral CD4[+] T cell repertoire of SuS patients.

Since spectratyping provides little quantitative detail about individual TCR clonotypes and sequence identity, we performed next-generation deep sequencing of the Vβ-CDR3 region[22,23] as a complementary method to study the TCR repertoire. This technology enables determination of clonal frequencies concomitant with sequence identification[24]. Clonal expansions in

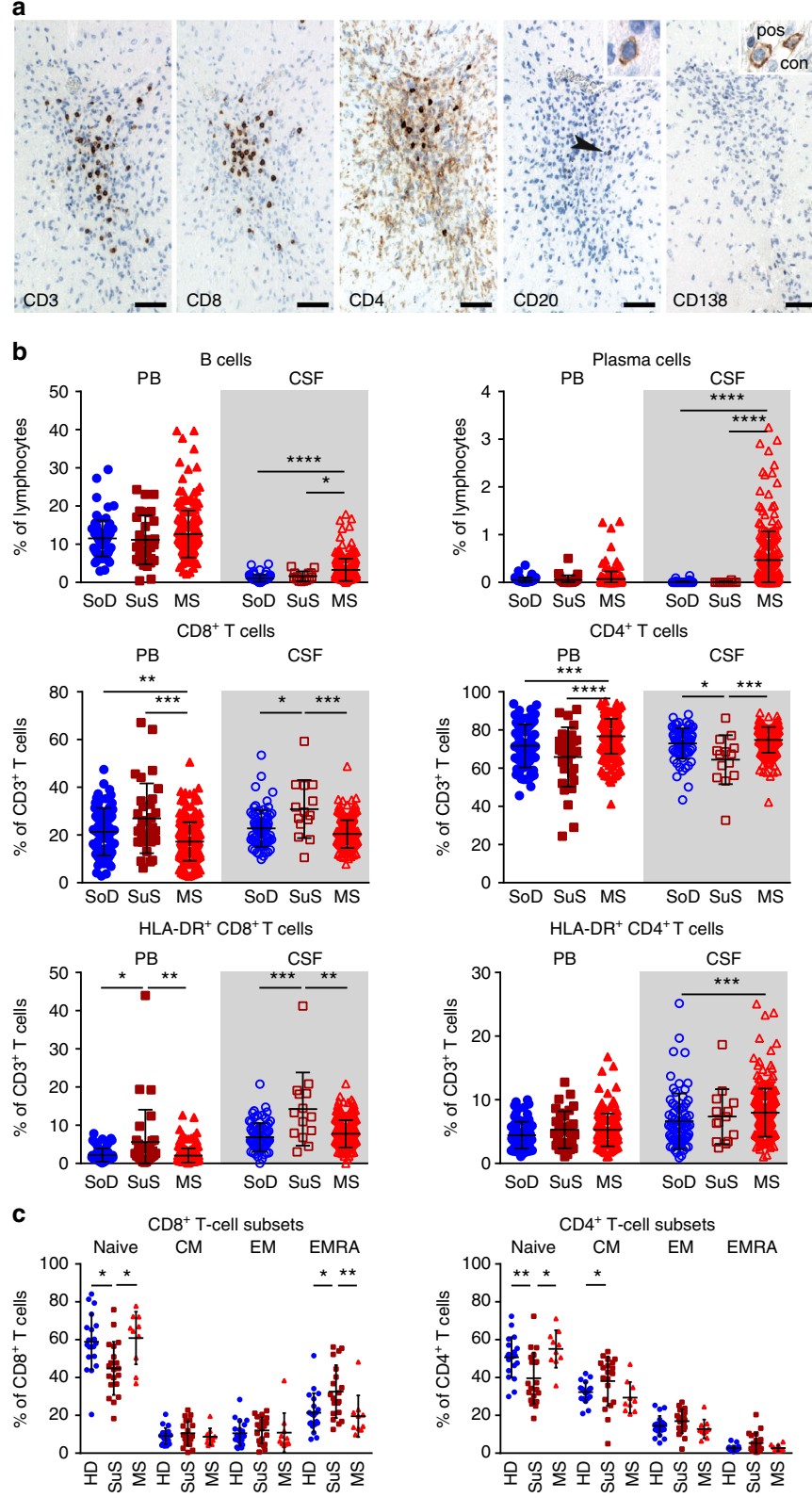

CD8$^+$ T cells were confirmed (Fig. 2b, c). Notably, clonality significantly correlated with proportions of CD8$^+$ T$_{EMRA}$ cells (Supplementary Fig. 4a) and accordingly highest clonality was observed in the CD8$^+$ T$_{EMRA}$ repertoire (Fig. 2b, c), both in treatment-naive and corticosteroid-treated patients (Supplementary Fig. 4b). Moreover, while the overall CD8$^+$ T cell repertoire was clonally diverse, the CD8$^+$ T$_{EMRA}$ repertoire was clearly restricted. Again, no clonal expansion was detected within the CD4$^+$ T cell compartment (Supplementary Fig. 4c). Interestingly, clinically active SuS patients showed significantly higher clonal expansion levels of CD8$^+$ T cells than patients in remission (Fig. 2d), further supporting the relationship to pathogenesis. These disease-related changes in clonality were not observed in the CD4$^+$ T cell compartment (Supplementary Fig. 4d).

**Fig. 1 Accumulation of activated CD8$^+$ T cells in SuS. a** Brain lesion from a patient with SuS stained for T cell markers CD3, CD8, and CD4, B cell marker CD20, and the plasma cell marker CD138. Bars: 50 μm. Most T cells belong to the CD8$^+$ T cell subset; light brown cells are most likely microglia cells; B cells are rare (arrowhead). One B cell is enlarged in the insert. CD138$^+$ plasma cells are absent from this and other lesions. The insert of the CD138 staining shows plasma cells from a positive control (Rasmussen encephalitis). **b** Graphs representing proportions of CD19$^+$ B cells (top left), CD138$^+$ plasma cells (top right) among lymphocytes and CD8$^+$ T cells (middle left), CD4$^+$ T cells (middle right), HLA-DR$^+$CD8$^+$ T cells (bottom left), and HLA-DR$^+$CD4$^+$ T cells (bottom right) among CD3$^+$ T cells in the peripheral blood (PB, closed symbols; SoD = 76; SuS = 32; MS = 227) and cerebrospinal fluid (CSF, open symbols; SoD = 76; SuS = 14; MS = 227) of somatoform disorders (SoD, blue circles), SuS (cayenne squares), and MS patients (red triangles up). **c** Quantification of naive, central memory (CM), effector memory (EM), and effector memory expressing CD45RA (EMRA) CD8$^+$ (left) and CD4$^+$ (right) T cell subsets in the peripheral blood of healhy donors (HD; n = 20, closed blue circles), SuS (n = 20, closed cayenne squares), and MS patients (n = 10, closed red triangles up). Statistical analysis was performed using Kruskal–Wallis test with Dunn's post-test. Error bars indicate the mean ± s.d.; p values: *p < 0.05; **p < 0.01; ***p < 0.001; ****p < 0.0001. Source data are provided as a Source Data file.

T cell responses to a given antigen are likely to involve clones that are private, i.e., only found in one individual, as well as public, i.e., shared by different individuals[25]. Especially in individuals that share at least some *HLA* background public clonal expansions might be directed against similar antigens. To further investigate the pathogenic relevance of clonally expanded CD8$^+$ T cells in SuS, we analyzed the 100 most prevalent clones in each patient and control. We identified 16 and 5 SuS-specific public clones in the total CD8$^+$ T cell and CD8$^+$ T$_{EMRA}$ repertoire, respectively, which were shared by at least two SuS patients, but absent in healthy individuals and MS patients (Table 1). These disease-specific public clones were not linked to other published disease-related clones, including known virus-specific clones[26–29].

Although the presence of public clonal T cell responses may suggest a shared specific pathogenic relevance[25], further analysis revealed that the ten clones with the highest copy number, which represented 20% of the total CD8$^+$ T cells and 55% of the CD8$^+$ T$_{EMRA}$ repertoire (Fig. 2e), were private and only found in individual SuS patients (Fig. 2f, Supplementary Table 3). Of note, SuS-specific private clones within the CD8$^+$ T$_{EMRA}$ repertoire exhibited unique characteristics with increased CDR3 length (Supplementary Fig. 4e, f) and higher numbers of nucleotide insertions in the N1 and N2 regions of the CDR3 (Supplementary Fig. 4g) when compared to public clones. In accordance with previous reports, CDR3 length is a prominent feature of private clones that is based on stochastic probability of a TCR recombination being more likely for a short CDR3 sequence[30].

Although the number of individuals was relatively small, SuS patients included in this analysis shared a similar *HLA* allele, except for one patient, who was homozygous for *HLA-C\*04* (Supplementary Table 4). Twelve out of 14 subjects expressed *HLA-C\*07*. Moreover, 5 out of 14 subjects expressed *HLA-C\*06*, which has a peptide-binding motif very similar to that of *HLA-C\*07*[31,32].

Taken together, analysis of the peripheral TCR repertoire in SuS patients demonstrates that SuS is not only characterized by a strong drive for differentiation of CD8$^+$ T cells into CD8$^+$ T$_{EMRA}$ cells but also by significant clonal expansion, particularly within CD8$^+$ T$_{EMRA}$ cells (but not in CD4$^+$ T cell compartment)—consistent with the hypothesis of an antigen-specific process underlying the disease pathogenesis.

**CD8$^+$ T cells exhibit increased cytolytic potential in SuS.** Given their clonal expansion and altered differentiation, we further characterized the function of CD8$^+$ T cells from SuS patients. Freshly isolated peripheral CD8$^+$ T cells from SuS patients expressed the cytotoxic molecules granzyme B (GrB) and perforin (Fig. 3a, Supplementary Fig. 1c) at higher levels than those of healthy individuals and MS patients, in line with increased proportions of CD57$^+$ bona fide CTLs in SuS (Supplementary Fig. 5a). Consistent with the immunophenotyping and repertoire

data, the highest increases of GrB and perforin were observed in the CD8$^+$ T$_{EMRA}$ cell subset (Supplementary Fig. 1c, Supplementary Fig. 5b), a phenomenon not explained by age-related changes (Supplementary Fig. 3a). Moreover, freshly isolated CD8$^+$ T cells of SuS patients released more cytotoxic granules upon TCR triggering in a re-directed lysis assay than those of healthy individuals and MS patients, suggesting a higher propensity for effector reactivity in SuS patients (Fig. 3b).

States of chronic inflammation are often characterized by dysfunctional regulatory cell populations[33,34] and/or resistance to immune regulation, e.g., by regulatory T cells (T$_{reg}$)[35,36]. Therefore, we assessed immune regulatory capacities in SuS. While frequencies (Supplementary Fig. 6a) and suppressive capacity (Fig. 3c) of T$_{reg}$ remained unaffected in SuS, CD8$^+$ T cells derived from SuS patients exhibited not only a high cytotoxic activity (Fig. 3a, b) but also resisted suppression by T$_{reg}$ (Fig. 3c). Of note, resistance to suppression by T$_{reg}$ was not observed in the CD4 T cell compartment (Supplementary Fig. 6b).

**CTLs accumulate at damaged CNS microvessels in SuS.** Quantitative and qualitative changes in circulating GrB- and perforin-expressing CD8$^+$ T cells suggested that CTLs could be responsible for SuS-specific endotheliopathy. We therefore investigated their contribution by immunohistochemistry of brain lesions from seven SuS patients (Supplementary Table 1, Table 2) We found that CD8$^+$ T cells significantly accumulated in the lumen and the perivascular space of brain microvessels of SuS patients (Fig. 4a). Unlike in encephalitis patients[37], CTLs did not penetrate deeply into the parenchyma. The proportion of CD8$^+$ T cells adhering to ECs was significantly increased in SuS patients compared to MS patients (Fig. 4m). Furthermore, GrB-expressing cells (Fig. 4b) were found in direct apposition to major histocompatibility complex (MHC) class-I-expressing brain ECs (Fig. 4c). Adherent CD8$^+$ T cells showed polarized GrB-containing cytotoxic granules indicating initiation of the cytolytic cascade (Fig. 4d, e). Polarization of cytotoxic granules in CD8$^+$ T cells was associated with apoptosis of ECs, as indicated by nuclear condensation (Fig. 4f, g) and TUNEL (terminal deoxynucleotidyl transferase-mediated dUTP-fluorescein nick end labeling) staining (Fig. 4h). Notably, the proportion of apoptotic EC was clearly increased in SuS compared to MS patients and non-inflammatory controls (Fig. 4n). Endothelial loss was associated with microhemorrhages (Fig. 4e, g) and leakage of the BBB as indicated by perivascular iron deposition (Fig. 4i). Staining for C9 neo showed the absence of complement deposition on ECs, suggesting that antibody-mediated complement activation does not play a role in EC degeneration (Table 2). In addition, small ischemic lesions, as indicated by degeneration of cerebellar granule cells, focal loss of astrocytes, myelin, and axonal damage, were present in three of the patients (Fig. 4j–l).

Together these data strongly support the hypothesis that an expanded and activated CTL-mediated endotheliopathy is a key

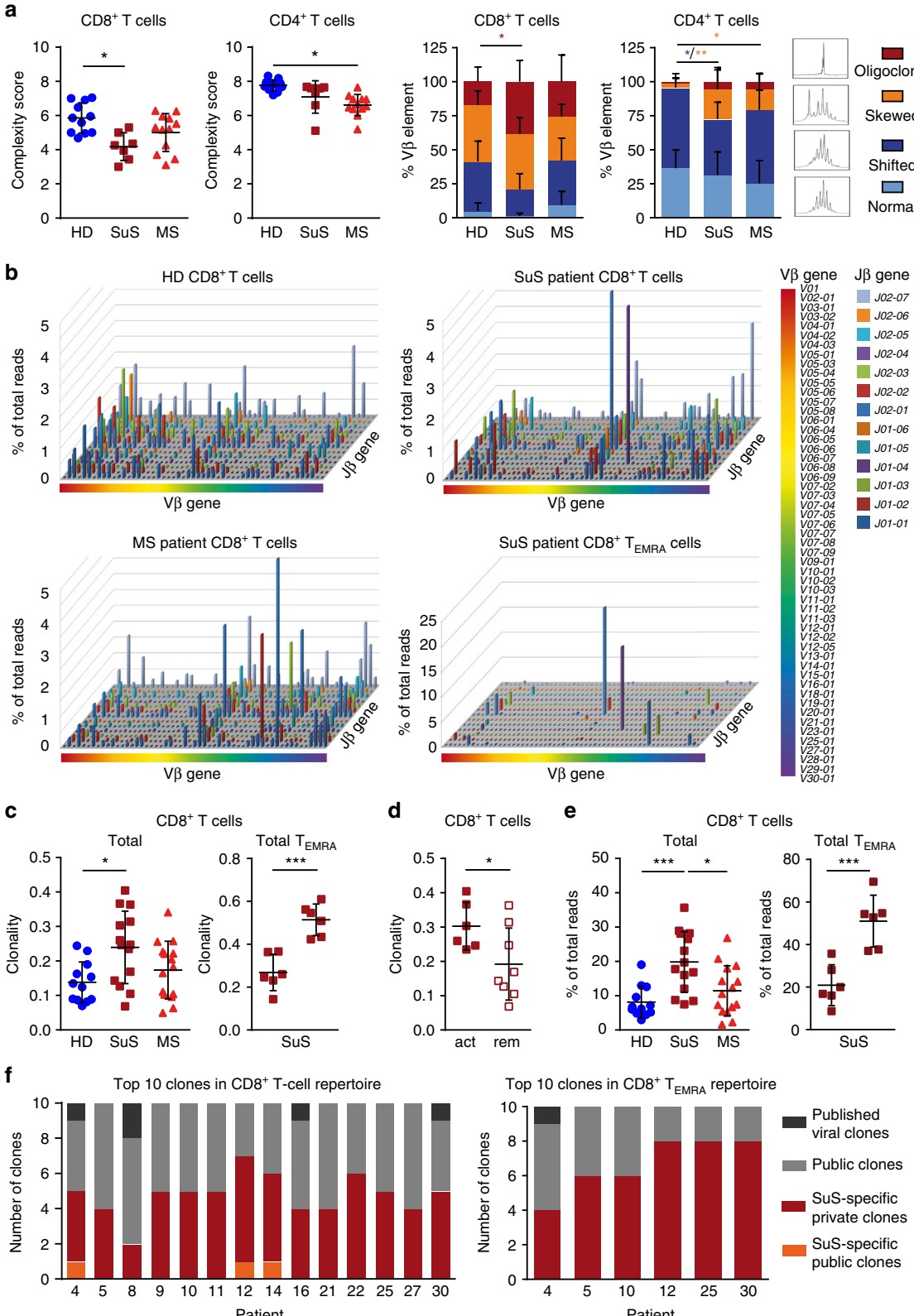

pathogenic feature associated with or even causative of SuS pathology.

**EC-HA⁺ mice as a model for CTL-mediated endotheliopathy.**
Based on the proposed antigen-driven and brain-endothelium directed pathogenesis in SuS, we developed a mouse model mimicking those features to directly test causal relationships. To this end, we generated (Slco1c1-CreER$^{T2}$ × Rosa26-Stop-HA)F1 mice, referred to as EC-HA⁺, (Supplementary Fig. 7a) that express the *influenza* virus hemagglutinin (HA), as an endothelial neo-antigen. Owing to the promoter used in this model, antigen expression was found in ECs of the brain and retina[38–40] as well as inner ear[41,42]—the target organs in SuS but not in other tested organs (Supplementary Fig. 7b). We have first assessed whether

**Fig. 2 Perturbations in the CD8$^+$ T$_{EMRA}$ repertoire of SuS patients. a** Complexity score (left) and CDR3 length distribution (right) of peripheral CD8$^+$ and CD4$^+$ T cell repertoires of HD (closed blue circles, $n = 11$), SuS (closed cayenne squares, $n = 7$), and MS patients (closed red triangles up, $n = 12$) assessed by TCR Vβ-CDR3 spectratyping. **b** Representative Manhattan plots showing TCR repertoires of CD8$^+$ T cells from a HD (top left), a SuS patient (top right, patient #10), a MS patient (bottom left), and SuS CD8$^+$ T$_{EMRA}$ cells (bottom right, patient #10). The x-axis depicts all the Vβ genes, the z-axis the Jβ genes, and the column heights represent the percentage reads for each V/J gene combination. **c** Clonality in the CD8$^+$ T cell repertoires of HD (closed blue circles, $n = 12$), SuS (closed cayenne squares, $n = 14$), and MS (closed red triangles up, $n = 15$) patients (left). Clonality in the total CD8$^+$ T cell and CD8$^+$ T$_{EMRA}$ repertoire of SuS ($n = 6$) patients (right). **d** Clonality of SuS patients with clinically active disease (closed cayenne squares, $n = 6$) or in clinical remission (open cayenne squares, $n = 8$). **e** Quantification of the ten most prevalent clones in the CD8$^+$ T cell repertoires of HD (closed blue circles, $n = 12$), SuS (closed cayenne squares, $n = 14$), and MS (closed red triangles up, $n = 15$) patients (left), as well as in the total CD8$^+$ T cell and CD8$^+$ T$_{EMRA}$ repertoire of SuS (closed cayenne squares, $n = 6$) patients (right). **f** Graph representing the proportion of viral, public, SuS-specific public, and SuS-specific private clones in the ten most prevalent clones in the total CD8$^+$ T cell (left) and T$_{EMRA}$ (right) repertoire of SuS patients. Statistical analysis was performed using Kruskal–Wallis test with Dunn's post-test (**a, c, e** left graph) or unpaired Student's $t$ test (**c** right graph; **d, e** right graph), respectively. Error bars indicate the mean ± s.d.; $p$ values: *$p < 0.05$; **$p < 0.01$; ***$p < 0.001$; ****$p < 0.0001$. Source data are provided as a Source Data file.

EC-HA$^+$ mice generate any immune reaction to tamoxifen-induced HA neoantigen. Therefore, prior to any CTL transfer, the CNS of tamoxifen-treated mice was analyzed by flow cytometry (five EC-HA$^+$ and five EC-HA$^-$ mice) and brain histology (three additional mice per group). No increased number of T cells and no T cell infiltration in different parts of the CNS (cortex, hippocampus, cerebellum, spinal cord, choroid plexus) were observed in EC-HA$^+$ animals. This indicates that the mere expression of a neoantigen by brain ECs is not sufficient for autoimmunity development. Adoptive transfer of activated HA-specific CTLs (Supplementary Fig. 8a, b) in EC-HA$^+$ or EC-HA$^-$ mice resulted in CD3$^+$ T cell infiltration in the retina, inner ear, and brain of EC-HA$^+$ but not in that of EC-HA$^-$ mice (Fig. 5a, representative sections and quantification), indicating that organ-specific antigen expression in ECs is responsible for T cell infiltration into the respective organs. Within the brain of EC-HA$^+$ mice distinct regions including the corpus callosum, hippocampus, cerebellum, and cortex were infiltrated in a time-dependent manner (Supplementary Fig. 8c). The vast majority of CNS-infiltrating T cells consisted of the adoptively transferred HA-specific CD45.1$^+$ CD8$^+$ T cells, whereas fewer endogenous CD45.1$^-$ CD4$^+$ and CD8$^+$ T cells were detected in the brain parenchyma (Supplementary Fig. 8a, d). CNS-infiltrating CD45.1$^+$ CTLs of EC-HA$^+$ mice exhibited CD107a surface exposure indicating cytotoxic activity (Fig. 5b). Furthermore, they also expressed pro-inflammatory cytokines including interferon (IFN)-γ and tumor necrosis factor (TNF)-α (Supplementary Fig. 8e). Adoptive transfer of antigen-specific CTLs not only resulted in T cell infiltrates in the brain, cerebellum, retina, and inner ear of EC-HA$^+$ mice but also led to significant weight loss and diminished motor performance when compared to EC-HA$^-$ mice (Fig. 5c). Neither clinical manifestations nor T cell infiltration was observed in EC-HA$^+$ mice upon transfer of naive HA-specific CD8$^+$ T cells (Supplementary Fig. 8d).

These data demonstrate that adoptively transferred antigen-specific CTLs are capable of inducing a neuroinflammatory disease in mice expressing the respective antigen in brain ECs as well as retina and inner ear, thus recapitulating major pathological features of SuS.

**CTL-mediated endotheliopathy results in a SuS-like pathology.** We next investigated the impact of antigen-specific CTLs on the microvessels of EC-HA$^+$ mice (Table 2). Following HA-specific CTL transfer, CTLs accumulated in CNS microvessels (Fig. 6a) as observed in SuS patients (Fig. 4a). Brain ECs from EC-HA$^+$ mice expressed MHC class-I molecules (Fig. 6b), permitting cognate interaction with HA-specific CD8$^+$ T cells. GrB-expressing CD8$^+$ T cells accumulated in the vessel lumen and parenchyma in close proximity to injured CD31$^+$ ECs in EC-HA$^+$ mice (Fig. 6c), similar to our observations in SuS patients (Fig. 4d, e). Notably,

GrB was polarized toward ECs, suggesting focal release of GrB by CTLs (Fig. 6d). HA expression in microvessels of the CNS resulted in endothelial injury following transfer of HA-specific CTLs, as indicated by apoptosis and loss of ECs (Fig. 6e–h, quantified in Fig. 6i). Both deposition of IgG and perivascular iron in the brain parenchyma further demonstrated breach of the BBB in EC-HA$^+$ mice as a result of adoptively transferred HA-specific CTLs (Fig. 6j). Small ischemic lesions were also detected in and above the corpus callosum 7 days after adoptive transfer of HA-specific CTLs into EC-HA$^+$ mice, as indicated by axonal damage, focal loss of both oligodendrocytes and astrocytes, and loss of endothelium (Fig. 6k). Notably, ECs from organs devoid of HA expression such as liver, heart, lung, and kidney remained intact (Supplementary Fig. 9).

In summary, our mouse model demonstrated that antigen-specific CTLs induce apoptosis of antigen-expressing ECs, resulting in microvascular damage to the brain cerebellum, retina, and inner ear, reproducing the pathological features of SuS. This also strongly supports the hypothesis that SuS is an antigen-specific CTL-driven endotheliopathy.

**α4 integrin blockade reduces disease in the mouse model.** To validate therapeutic targets using our preclinical model, EC-HA$^+$ mice were treated with an anti-α4 integrin monoclonal antibody (mAb) that impedes the interaction of VLA-4-expressing T cells with vascular cell adhesion molecule 1 (VCAM-1) expressed on ECs, thereby inhibiting adhesion of lymphocytes to the endothelium[43,44] and subsequent transmigration across the BBB[19,45]. Antibody blockade of α4 integrin resulted in marked disease amelioration, as indicated by stable weight and preserved motor performance in EC-HA$^+$ mice compared to their counterparts treated with the corresponding isotype control (Fig. 7a). Accordingly, the numbers of CNS-infiltrating endogenous T cells as well as adoptively transferred HA-specific CTLs were significantly decreased in anti-α4 integrin-treated mice (Fig. 7b). These data thereby identify a therapeutic strategy to interfere with the CTL-mediated endotheliopathy that drives the SuS-like phenotype in the mouse model.

**Disease severity decreased in SuS patients under natalizumab.** Based on the assumption that immune cell trafficking into the CNS and immune–endothelium interactions are crucial components of SuS pathology, and following the strong support of this hypothesis in our preclinical model, we used natalizumab, a licensed humanized mAb approved for the therapy of active MS and directed against the α4 integrin[19,46]. Natalizumab could prevent binding of SuS-patient derived CD8$^+$ T cells onto a confluent human brain microvascular endothelial cell (HBMEC) monolayer under shear flow conditions in vitro (Fig. 7c).

**Table 1 SuS-specific public CD8+ T cell and CD8+ T_EMRA clones.**

| SuS-specific amino acid sequence | Vβ gene | Jβ gene | 4 | 5 | 8 | 9 | 10 | 11 | 12 | 14 | 16 | 21 | 22 | 25 | 27 | 30 |
|---|---|---|---|---|---|---|---|---|---|---|---|---|---|---|---|---|
| **CD8+ T cells** | | | | | | | | | | | | | | | | |
| CSVPGLDYTF | V29-01 | J01-02 | | | X | | | X | | | | X | | | | X |
| CASSQERSGGSSGANVLTF | V04-01 | J02-06 | X | | | | | | | | | | | | | |
| CATSDGTRVYEQYF | V27-01 | J02-01 | | X | | | | | | X | | | X | | | |
| CASSWGGFNNEQFF | V27-01 | J02-01 | | X | | | | | | | | | X | | | |
| CASSLEGRDSHFGANLTF | V07-09 | J02-06 | X | | | | | | | X | | | | | | |
| CASSLNERDNQPQHF | V05-06 | J01-05 | | | | | | | | X | | | | | | |
| CASSIPARKDTEAFF | V04-02 | J01-01 | | | | X | | | | | | | | | X | |
| CSVVGTGFSYEQYF | V29-01 | J02-07 | | | | | | | X | | | | | | | |
| CASSVGGAGFQPQHF | V19-01 | J01-05 | | | X | | | | | X | | | X | | | |
| CASSQDFSSGQPQHF | V04-03 | J01-05 | | X | | | X | | | | | | | X | | |
| CASTSKRGGYNEQFF | V06 | J02-01 | | | | | X | | | | | | | | | |
| CASSIAASGANEQFF | V19-01 | J02-01 | | | | | X | | | | | | | | | X |
| CASGTGWHEQYF | V25-01 | J02-07 | | | | | | | | | X | | | X | | |
| CASSFGSTNIQYF | V05-04 | J02-04 | | | | X | | | | X | | | | | X | |
| CASSNGQGANEQFF | V06-05 | J02-01 | | | | | | X | | | | | | | X | |
| CSANTGVEQYF | V20-01 | J02-07 | | | | | X | | | | | | | | | |
| **CD8 T_EMRA** | | | | | | | | | | | | | | | | |
| CASLGTGGMETQYF | V10-02*01 | J02-05*01 | | X | NA | NA | NA | NA | X | NA | NA | NA | NA | | | |
| CASRIGPGNNEQFF | V27-01*01 | J02-01*01 | X | | NA | NA | NA | NA | | NA | NA | NA | NA | | | |
| CASSSPDRSYNSPLHF | V07-09 | J01-06*01 | X | | NA | NA | NA | NA | | NA | NA | NA | NA | | | |
| CASSPIPGQLGAGELFF | V28-01*01 | J02-02*01 | X | | NA | NA | NA | NA | X | NA | NA | NA | NA | X | | |
| CASSQDYDSNYGYTF | V04-03*01 | J01-02*01 | | | NA | NA | NA | NA | | NA | NA | NA | NA | | | X |

Table summarizing SuS-specific public CD8+ T cell clones and CD8 T_EMRA clones shared by ≥2 SuS patients and absent in HD (n = 12) and MS patients (n = 15) as well as 1052 published viral clones and other published disease-related sequences curated in the immunoACCESS™ data base. Vβ gene and Jβ gene usage are shown
NA not analyzed

Encouraged by these findings, we treated off-label four SuS patients with a progressive relapsing disease course and who did not respond to prior immune therapies. Notably, the four patients treated with natalizumab seemed to have reduced relapse rate and disease progression (Fig. 7d, e), as far as the limited pretreatment time windows in two of the patients permit the conclusion. Calculation and visualization were performed in analogy to Cree et al.[47]. Moreover, in two instances natalizumab discontinuation was followed by relapses (Fig. 7d). As an example, a reduction in CNS inflammation was demonstrated on magnetic resonance imaging (MRI) with fewer SuS characteristic "snowball-like" lesions[48] in the corpus callosum of a SuS patient, 15 months after initiating natalizumab treatment (Fig. 7f). Despite all caution related to the off-label use of natalizumab in only four patients, this suggests that VLA-4 blockade attenuates pathogenic immune cell trafficking but, given the relapses upon cessation, probably does not silence the underlying immune process.

Thus, using anti-α4 integrin as a therapeutic approach corroborates our hypothesis that CD8+ T cells promote the development of an endotheliopathy in SuS and are a potential therapeutic target for such pathology.

## Discussion

Our study suggests that SuS is a CTL-driven endotheliopathy against unidentified antigen(s). In-depth functional immune phenotyping and TCR profiling of blood and CSF lymphocytes from patients with SuS revealed CTLs, specifically terminally differentiated CD8+ T_EMRA cells, to be putative actors in its pathogenesis. The pathology of SuS brain lesions provides strong evidence for a cytolytic endotheliopathy, in which CD8+ T cells recognize an unknown antigen on ECs, resulting in endothelial damage, small ischemic foci, and microhemorrhages. Using a reverse translational approach, we directly tested for causation in vivo between EC-targeting CTLs and the main neuropathological features of SuS in a mouse model. A similar pathology could be induced experimentally in this mouse model, in which CTLs target a neo-antigen expressed in ECs of the brain, retina, and inner ear, using a promoter expressed in ECs exhibiting tight junctions[38–40,49]. However, the phenotypes may not be exclusively linked to T cell engagement of BBB ECs as the Slco1c1 promoter can be active in choroid plexus and some CNS-resident cells[39]. Disease amelioration was achieved in the mouse model by blocking adhesion and trafficking of CTLs using an anti-α4 integrin mAb. Finally, we translated these findings back to the clinic by treating four SuS patients with natalizumab. Partial attenuation of inflammatory activity in these patients further corroborates the role of T cell/EC contact in SuS pathogenesis.

We investigated the pathophysiological processes underlying SuS to gain further insight into immune cell/EC interactions resulting in BBB dysfunction. A recent study revealed that murine MOG-specific CD4+ T cells increase the BBB permeability in an IFN-γ, GrB, and contact-dependent manner indicating a causative role of T cells in BBB dysfunction[50]. However, the particular role of CD8+ T cells in brain endothelium damage remains mostly unknown. Collectively, although the contribution of CD4+ T cells remains to be investigated, our results argue that SuS is a CD8+ T cell-driven microvascular endotheliopathy associated with neuro-inflammation (Fig. 8). It is therefore tempting to speculate that circulating, activated, clonally expanded CD8+ T_EMRA cells accumulate in CNS microvessels, where they are involved in the induction of EC apoptosis. Endothelial injury may result in vascular leakage and lead to the formation of small ischemic lesions predominantly in the corpus callosum, cerebellum, retina, and inner ear.

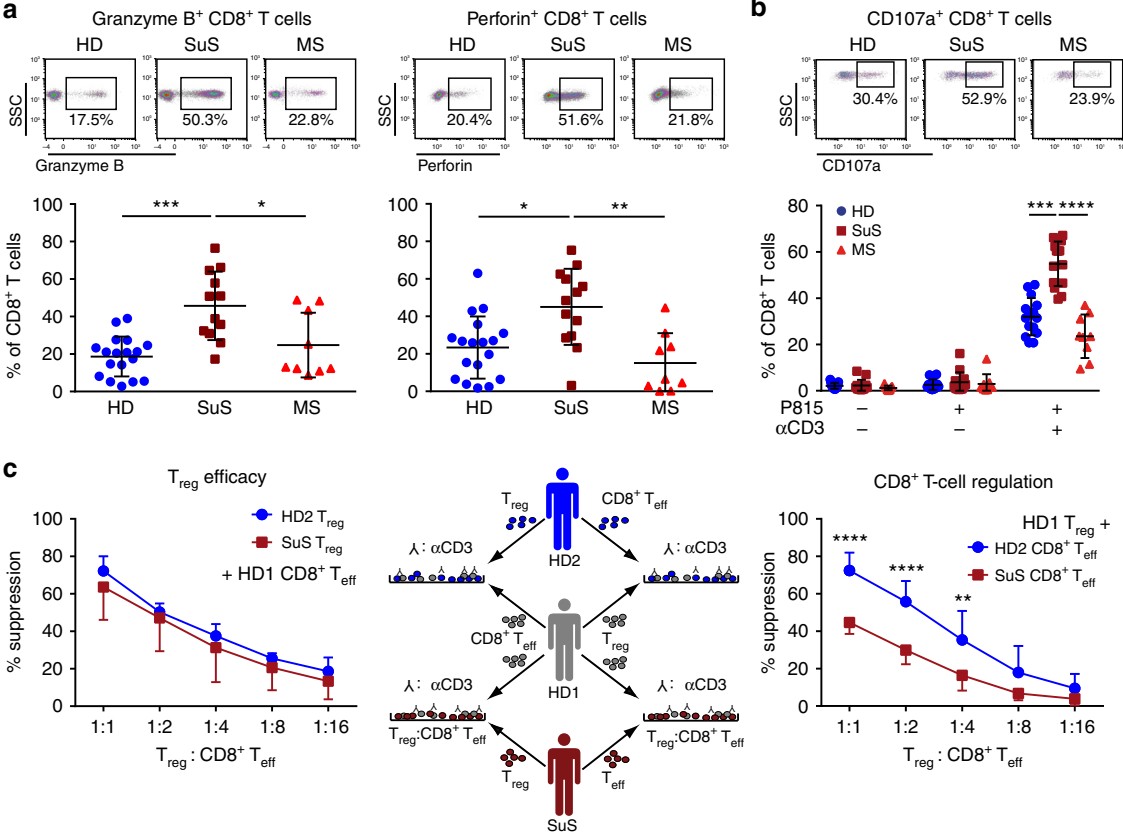

**Fig. 3 CTLs are more prevalent in the periphery of SuS patients. a** Representative examples (top) and quantification (bottom) of granzyme B (GrB; left) and perforin (right) expressing CD8+ T cells circulating in the blood of HD (closed blue circles, $n = 18$), SuS patients (closed cayenne squares, $n = 13$), and MS patients (closed red triangles up, $n = 9$). **b** Representative examples (top) and quantification (bottom) of CD107a-expressing CD8+ T cells from HD (closed blue circles, $n = 15$), SuS patients (closed cayenne squares, $n = 14$), and MS patients (closed red triangles up, $n = 9$) under the indicated conditions of re-directed lysis assay with αCD3-linked P815 cells. **c** Middle: Experimental set-up of allogenic suppression assays to test the suppressive capacity of regulatory T cells. Left: Graphs representing the degree of suppression of proliferation of HD CD8+ T cells upon co-culture with titrated numbers of $T_{reg}$ from HD (closed blue circles) and SuS (closed cayenne squares) ($n = 3$). Right: Graphs representing the degree of suppression of HD (closed blue circles, $n = 7$) and SuS patient (closed cayenne squares, $n = 7$) CD8+ T cell proliferation upon co-culture with titrated numbers of independent HD $T_{reg}$. Statistical analysis was performed using Kruskal–Wallis test with Dunn's post-test (**a, b**) or two-way ANOVA with Bonferroni post-test (**c**), respectively. Error bars indicate the mean ± s.d.; $p$ values: *$p < 0.05$; **$p < 0.01$; ***$p < 0.001$; ****$p < 0.0001$. Source data are provided as a Source Data file.

CD8+ T cells participate in the endotheliopathy that is a hallmark of experimental cerebral malaria[51,52] suggesting that CD8+ T cell-mediated induction of BBB leakage may be a general mechanism also occurring in other neuro-infectious (e.g., viral, toxoplasmosis)[53,54] or putative autoimmune neuro-inflammatory diseases (e.g., Rasmussen encephalitis[55]). While our findings point toward key mediators of cytotoxic vascular breakdown, recent work in an animal model of cerebral malaria elegantly demonstrated that the vascular breakdown associated with alterations in tight junction protein expression is predominantly mediated by IFN-γ[52]. Furthermore, alterations in tight junctions were also observed in a natural feline model for LGI1 encephalitis where the ECs stay intact, but tight junctions are lost. While many studies focus on the mechanisms of transmigration and trafficking into the brain, ECs have rarely been considered as a target of antigen-specific immune recognition or BBB breakdown a consequence of an endothelial-directed immune attack in neuro-inflammation. Of note, while apoptosis of brain ECs was consistent and marked in SuS, it was mostly absent in MS patients indicating that in SuS CD8+ T cells attack the endothelium, whereas in MS they mainly transmigrate across the BBB. Our histological data clearly showed attachment of CTLs to brain ECs raising the question how CD8+ T cells mediate attachment to the microvessels despite high shear force. Collectively, our data argue

for a plausible scenario in which VLA4/VCAM-1 binding enables physical interaction between CD8+ T cells and brain ECs. In turn, this interaction facilitates TCR recognition of MHC:peptide complexes on the luminal side of ECs. The subsequent release of IFN-γ may further enhance VCAM-1 and MHC class-I expression on ECs, facilitating the cognate interactions between antigen-specific CD8+ T cells and ECs. Under shear flow conditions, primed T cells have been shown to adhere stronger to brain ECs than their naive counterparts[56]. However, further studies are warranted to identify the molecular mechanisms used by CD8+ T cells in different neuro-inflammatory diseases to create BBB leakage and promote brain pathology.

Our study mainly focused on the pathophysiological role of CD8+ T cells on ECs in the context of SuS. As previously shown[9], and confirmed here, CD8+ T cells (and other immune cells) infiltrate the CNS parenchyma of SuS patients, where they cause microhemorrhages and neural tissue destruction. Activation, clonal expansion, and differentiation into CD8+ $T_{EMRA}$ cells[20,57,58] indicate priming of CD8+ T cells with cognate antigen(s) presented on MHC class-I molecules as a crucial pathogenic step in SuS. T cell responses to a single antigen can be orchestrated by different T cells exhibiting specificities to distinct antigenic epitopes resulting in multiple disease-specific clones within the TCR repertoire. T cell responses to specific antigens

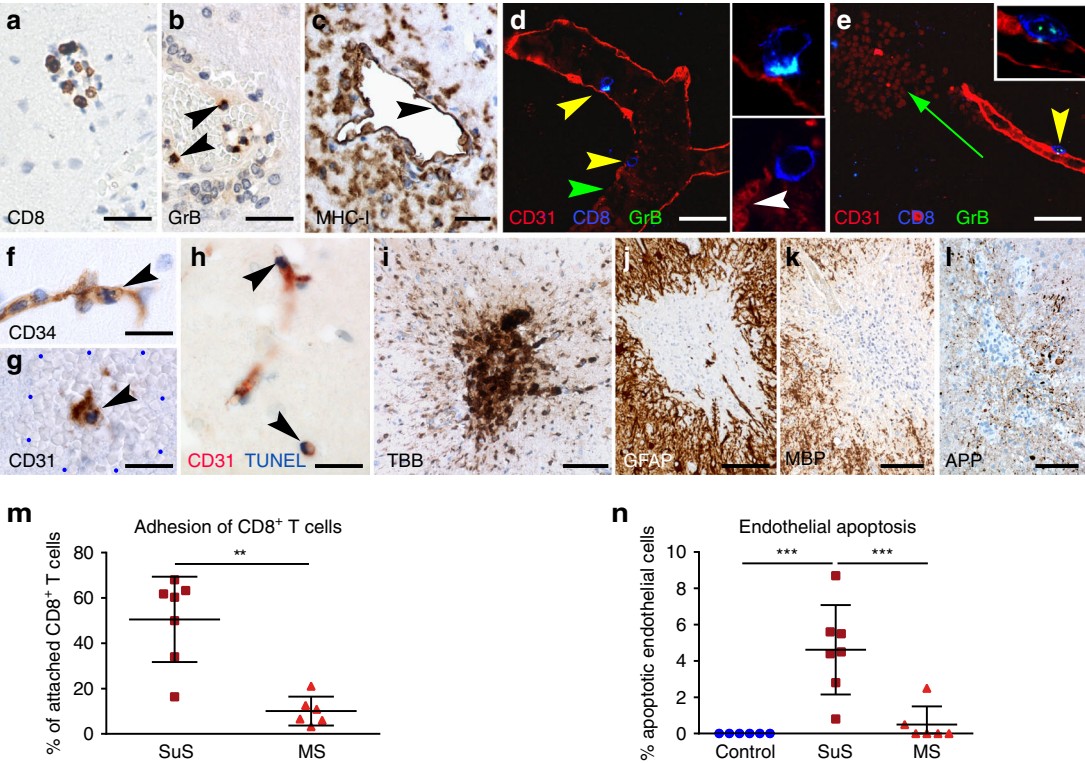

**Fig. 4 CTLs accumulate in damaged microvessels of SuS patients CNS biopsies of SuS patients ($n = 7$). a** Accumulation of CD8+ T cells (brown) in a brain microvessel. **b** GrB+ cells in a brain microvessel; arrowheads indicate GrB+ cells attached to the vessel wall. **c** MHC class-I expression on ECs (arrowhead). Several leukocytes adhere to the endothelium. **d** Yellow arrowheads point to CD8+ T cells (blue) attached to CD31+ ECs (red). The enlargement of the upper CD8+ T cell shows its GrB+ (green) granules polarized toward the ECs. Owing to close proximity of the green GrB+ granules to the blue CD8+ cell membrane, these granules have a cyan blue color. The lower CD8+ T cell is migrating through the damaged (white arrowhead) vessel wall. **e** The same triple staining for GrB (green), CD8 (blue), and CD31 (red) shows a cytotoxic T cell attached to an intact vessel wall (yellow arrowhead). The upper left corner shows a microhemorrhage (green arrow) with parenchymal erythrocytes. **f** An apoptotic CD31+ EC with a condensed nucleus (arrowhead) within a microhemorrhage. **g** Staining for CD34 shows an EC with an apoptotic condensed (arrowhead) nucleus within a microhemorrhage encircled by the blue dots. **h** TUNEL-positive nuclei (arrowheads) of CD31+ ECs. **i** Turnbull blue (TBB) shows iron deposition around a blood vessel. **j–l** Small ischemic lesion combining loss of GFAP+ astrocytes, MBP+ myelin, and axonal damage shown by APP+ spheroids. **m** Proportions of CD8+ T cells in contact with ECs in brain specimen of SuS (cayenne squares, $n = 7$) and MS patients (red triangles up, $n = 6$). **n** Quantification of CD34+ endothelial apoptosis in brain specimen derived from non-inflammatory controls (blue circles, $n = 6$), SuS (cayenne squares, $n = 7$), and MS patients (red triangles up, $n = 6$). Scale bars: 20 μm (**a**, **b**, **f–i**), 25 μm (**c–e**), 200 μm (**j**, **l**). Statistical analysis was performed using Mann–Whitney test (**m**) or one-way ANOVA with Bonferroni post-test (**n**), respectively. Error bars indicate the mean ± s.d.; $p$ values: **$p < 0.01$; ***$p < 0.001$. Source data are provided as a Source Data file.

are likely to involve clones that are public and shared by individuals with response to the same MHC:peptide complex[25]. A recent study by Zhao et al. suggested that autoimmunity is mainly driven by public clones[59]. Here we found SuS-specific public clones that were shared among at least two SuS patients and absent in both published common viral and clones expanded in other diseases[26–29]. These findings further support the idea that an antigen-dependent process, with similar antigen(s) across different individuals suffering from the disease, is a prerequisite for the disease. This hypothesis was further strengthened by our observation that clonality of CD8+ T cells is also associated with the disease course, similar to another potentially CD8+ T cell-driven disease, Rasmussen encephalitis[24]. Memory subsets such as CD8+ $T_{EM}$[60] and CD8+ $T_{EMRA}$ cells are known to comprise the majority of antigen-primed CD8+ T cells, and accordingly, we observed the highest degree of clonal expansion within the CD8+ $T_{EMRA}$ subset.

Further support for the possibility of public clones perpetuating the disease comes from HLA analysis. Except for one SuS patient who was homozygous for *HLA-C*04*, all SuS patients expressed *HLA-C*06* and/or *HLA-C*07*. Comparing the

peptide-binding motifs of these HLA-C allotypes revealed that the binding motif of *HLA-C*06:02* and *HLA-C*07:02* are almost identical. The homology of the HLA–peptide binding for these two allotypes has been recently proven in a study clustering distinct *HLA-C* allotypes according to predictive binding motifs[32]. Cluster analysis indicated that *HLA-C*06:02* and *HLA-C*07:02* bind the cognate peptide and cluster together in close proximity to *HLA-C*04*[32]. Since we identified identical clones in 3 out of 12 patients who share *HLA-C*07*, it is tempting to speculate that these disease-specific public clones might recognize the same antigen/HLA complex. Thus a similar genetic background might evoke similar immunological mechanisms that shape the CD8+ T cell-mediated autoimmunity. However, most of the SuS-specific clones found in the CD8+ $T_{EMRA}$ repertoire were private to individual SuS patients. Although this might hint toward recognition of different antigens, it does not exclude that some of them might also recognize similar antigens, because HLA-peptide recognition by TCRs[61] is highly degenerate, i.e., a given HLA–peptide complex can be recognized by structurally diverse TCRs. Furthermore, such clones may have developed owing to epitope spreading in the course of the disease.

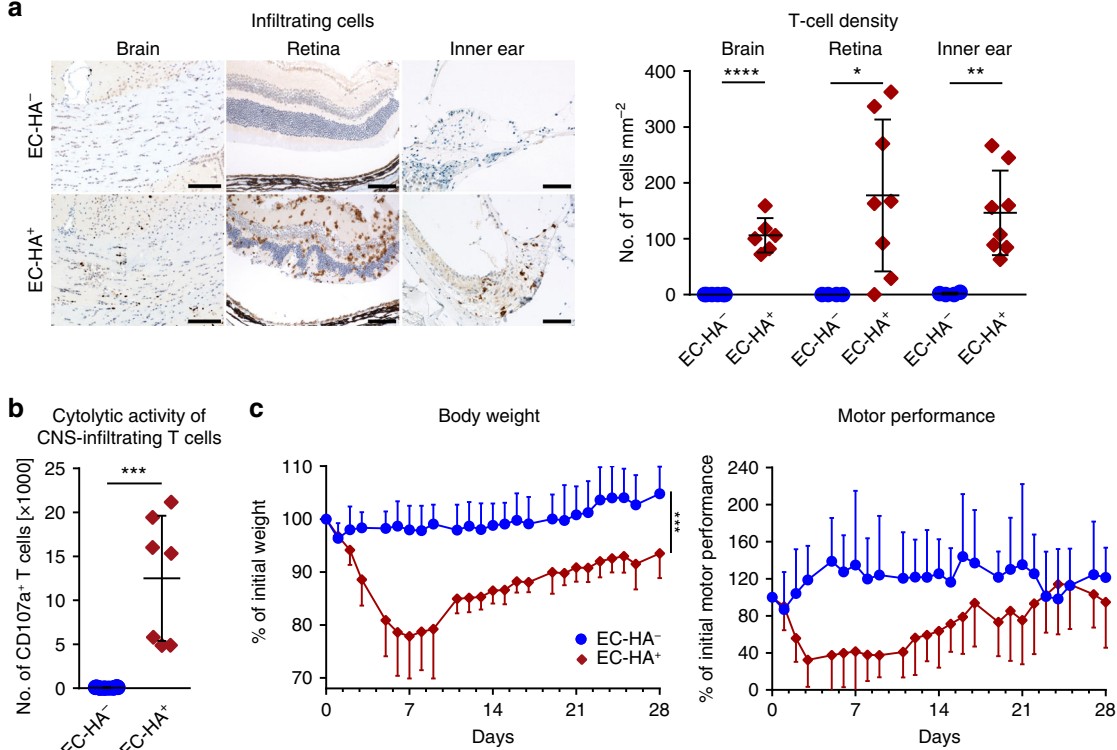

**Fig. 5 EC-HA$^+$ mice as an in vivo model for CTL-mediated endotheliopathy. a** Investigation of T cell CNS infiltration in the brain, retina, and inner ear of EC-HA$^-$ or EC-HA$^+$ recipient mice on day 7 after adoptive transfer. Left: Representative histological sections documenting T cell infiltration (CD3$^+$, brown). Right: Quantification of tissue-infiltrating T cells in EC-HA$^-$ (closed blue circles) or EC-HA$^+$ (closed cayenne diamonds) recipient mice ($n = 4$–8 per group); data for retina and inner ear originate from 2 independent experiments, involving transfer of cytotoxic CD8$^+$ T cells alone or with control IgG. **b** Quantification of the cytolytic activity of CNS-infiltrating T cells analyzed as CD107a expression on day 7 after adoptive transfer in CD45.2$^+$ EC-HA$^-$ (closed blue circles) or EC-HA$^+$ recipient mice (closed cayenne diamonds) ($n = 7$–8 per group, 2 independent experiments). **c** Weight loss (left; 11–12 mice per group) and rotarod motor performance (right; 7–8 mice per group) following adoptive transfer of HA-specific CTLs in EC-HA$^-$ (closed blue circles) or EC-HA$^+$ (closed cayenne diamonds) mice. Data are from three (body weight) or two (motor performance) independent experiments. Statistical analysis was using the unpaired Student's $t$ test (**a**, **b**) or two-way ANOVA with Bonferroni post-test (**c**), respectively. Error bars indicate the mean ± s.d.; $p$ values: *$p < 0.05$; **$p < 0.01$; ***$p < 0.001$; ****$p < 0.0001$. Source data are provided as a Source Data file.

Why would brain ECs act as a target for CD8$^+$ T cell recognition? There are different possibilities, although the nature of the targeted antigen(s) is currently unknown. One hypothesis is virus infection that results in the presentation of viral-derived peptides on the endothelium of the affected organs and/or in upregulation of HLA class-I expression presenting viral or self-peptides on ECs. For example, a study by Petty et al. showed that patients with retinocochleocerebral vasculopathy had a previous cytomegalovirus (CMV) infection. This raises the question as to how CD8$^+$ T cells of SuS patients circumvent common resistance mechanism of ECs toward CTL-mediated attack. A recent study has shown that upregulation of programmed death ligand 1 on ECs during systemic lymphocytic choriomeningitis mammarenavirus infection inhibits killing of infected ECs by virus-specific CTLs[62], and further studies are warranted to reveal the underlying mechanisms resulting in an antigen-specific lysis despite these protective mechanisms, as shown in the current study. Although CMV-derived antigens also induce CD8$^+$ T$_{EMRA}$ differentiation[63,64], the absence of CMV-related clones (except for one patient) among the top ten expanded clones of the CD8$^+$ T$_{EMRA}$ repertoire of SuS patients strongly suggests that CMV, a dominant member of the herpes virus family, is not the driving force in this disease. This was further corroborated by our finding that we could not find CMV infection of ECs in our SuS cases. An alternative hypothesis is a CD8$^+$ T cell-mediated autoimmune reaction against endothelium-specific antigen(s). The existence of SuS-specific public and private clones that were absent in 1052 published viral clones favors the hypothesis of presentation of endogenous peptides by ECs. Additional experiments, e.g., using endothelium-derived antigen libraries, might provide further support for this hypothesis.

Our mouse model clearly demonstrated that ECs are primarily attacked and killed in an antigen-specific processes mediated by CTLs, when the respective antigen is presented in the context of MHC class-I molecules on the endothelium. Whether disease-specific antigen expression on microvascular ECs of SuS patients also induces killing by CD8$^+$ T cells needs to be further elucidated. However, existence of AECA in some SuS patients[12,13,65] points in this direction. Influence of CD8 T cells might be transient, leading to functional disturbance of the BBB, or permanent, leading to destruction of EC, as demonstrated here.

Finally, as a proof of principle, treatment of SuS patients with natalizumab, a humanized mAb directed toward the α4 integrin (CD49d) that prevents binding of VLA-4-expressing lymphocytes to VCAM-1 expressed on inflamed ECs[44,46,66], was used to demonstrate the potential impact of a non-antigen-specific displacement of CD8$^+$ T cells from microvessels of the affected organs as means to ameliorate the disease, thereby indirectly corroborating our hypothesis of the direct involvement of these T cells in endothelial injury. Although the number of treated

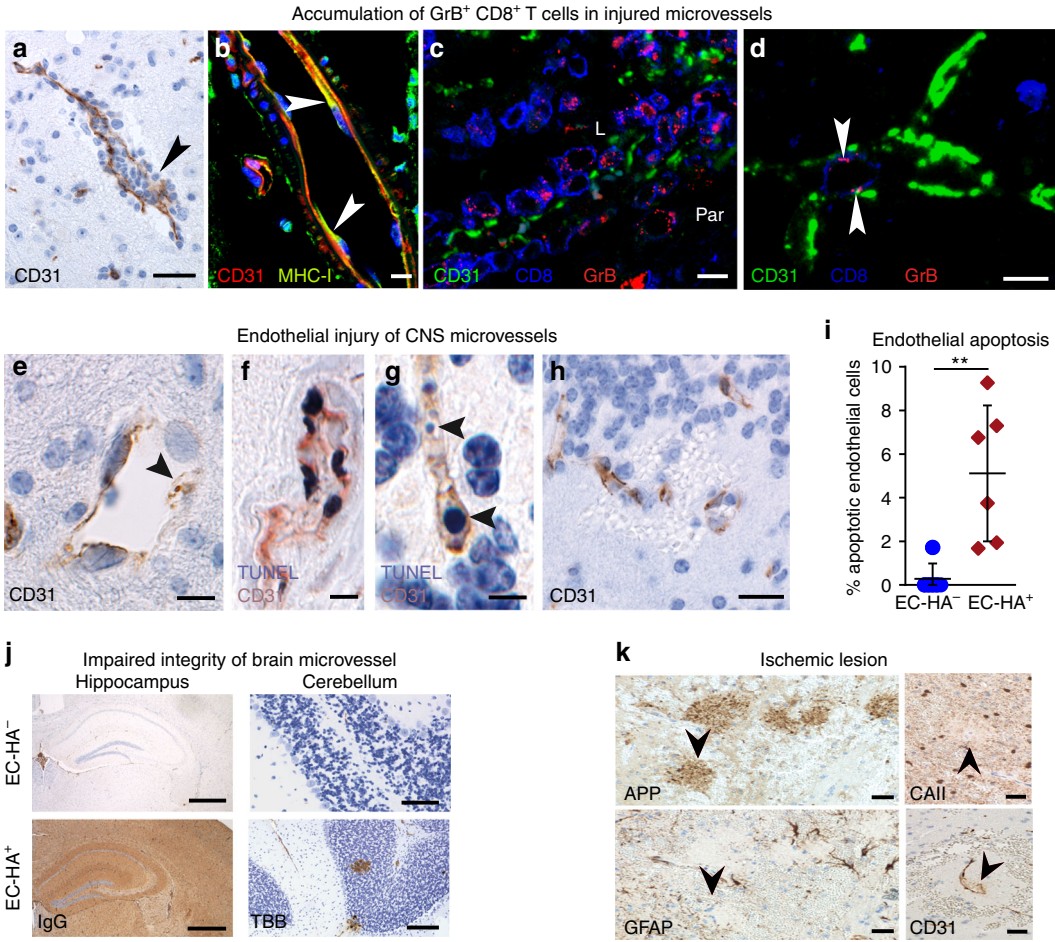

**Fig. 6 CTLs accumulate in damaged microvessels of EC-HA+ mice.** Representative brain images of EC-HA+ mice 4 days after adoptive transfer of cytotoxic CD8+ T cells (n = 6). **a** Accumulation of leukocytes in CNS microvessels. **b** Expression of MHC class-I molecules (green) on CD31+ (red) ECs; nuclei marked with TO-PRO-3 (blue). **c** Presence of GrB+ (red) CD8+ (blue) T cells in vessel lumen (L) of a battered CD31+ (green) blood vessel. Par parenchyma **d** GrB+ granules (red, arrowheads) CD8+ cells (blue) in close contact with the lumen of a small CD31+ blood vessel (green). **e** Representative image exhibiting CD31+ (brown) endothelium loss (arrowheads) in CNS microvessels of EC-HA+ mice following CD8+ T cell transfer. **f** Apoptosis and **g** DNA fragmentation (arrowheads) of CD31+ ECs (brown) detected with TUNEL staining (black). **h** Microhemorrhages in close proximity to destroyed vessels. Scale bars 10 μm. **i** Percentage of endothelial apoptosis assessed condensed nuclei in CD34+ endothelial cells (74–206 cells per mouse) of EC-HA− (closed blue circles) or EC-HA+ mice (closed cayenne diamonds) (n = 6 per group). Scale bars 100 μm. Statistical analysis was performed by unpaired Student's t test. Error bars indicate the mean ± s.d. p value: **p < 0.01. Source data are provided as a Source Data file. **j** Deposition of IgG in the hippocampus and turnbull blue staining (TBB) of the cerebellum of EC-HA− and EC-HA+ mice after CTL transfer. Scale bars 500 μm. Data from two independent experiments, 5–6 mice per genotype are shown. **k** Representative small ischemic lesion in and above the corpus callosum at day 7 post-transfer in an EC-HA+ mouse. APP labeling (arrowhead) reveals axonal damage. Focal loss of astrocytes is visualized by the absence of GFAP (arrowhead). Detection of carbonic anhydrase II (CAII) reveals the focal loss of oligodendrocytes (arrowhead). The CD31 staining shows loss of ECs (arrowhead). Nuclear counterstaining was performed with hematoxylin (blue) (n = 6 per group).

patients is too limited to draw firm conclusions, and despite one previously reported case of putative SuS not benefiting from this therapy[67], treatment with natalizumab along with other therapy was associated with a reduced clinical and paraclinical activity and fewer relapses in the four treated patients. Relapse activity occurring after discontinuation of the natalizumab treatment in two patients not only supports the putatively beneficial role of this drug in SuS but also suggests that it does not resolve the underlying condition. Thus natalizumab may offer a more targeted and effective treatment regimen in addition to the current treatment recommendations[68] and even more generally for diseases where CD8+ T cell-mediated degeneration can be considered a key element (e.g., Rasmussen encephalitis[69]). Depending on the disease, treatment with efalizumab (a humanized anti-LFA-1 antibody) that also prevents binding of lymphocytes to the endothelium may be an alternative to natalizumab. Along these lines, a recent study demonstrated that anti-LFA1 antibody treatment resulted in disease amelioration in the animal model of experimental cerebral malaria by a mechanism displacing CD8+ T cells from cerebrovascular ECs[52]. Nevertheless, further studies are required to elucidate which of these treatments is the most efficacious. For that purpose, the SuS-like mouse model developed in this study will be valuable.

Our study highlights a potential fruitful avenue of research focusing on the role of CD8+ T cells recognizing and interacting with brain endothelium. Given that this mechanism is relevant to several neuroinflammatory and neuroinfectious diseases, further characterization of the nature of the target antigen(s), the reactivity patterns of ECs under physiological and pathological conditions, and approaches to interfere with this key pathogenic step, potentially using this animal model, are important next steps to follow.

**Table 2 Neuropathological evaluation of SuS patients and Slco1c1-HA/EC-HA mice.**

| | SuS | Slco1c1-HA/EC-HA mice |
|---|---|---|
| **Frequent location of lesions** | | |
| | Leptomeninges[a,b,c] | Leptomeninges |
| | ND | Choroid plexus |
| | Snow ball lesions in corpus callosum[a,b,c] | Corpus callosum |
| | Subcortex[a,b,c] | Subcortex |
| | Cerebellum[a,b,c] | Cerebellum |
| | ND | Spinal cord |
| | Retina[b,c] | Retina |
| | Inner ear[b] | Inner ear |
| **Target cells** | Brain ECs[a,b,c] | Brain ECs |
| **Neuropathological changes** | | |
| | Binding of CTLs to luminal side of BVs, invasion into vessel walls[a], moderate parenchymal infiltration of CD8+ T cells, rare number of CD20 B cells, and the absence of plasma cells[a,b,c] | Binding of CTLs to luminal side of BVs, parenchymal infiltration of CD8+ T cells |
| | Occlusion of BVs, swollen ECs, thickened vessel walls[b,c], apoptosis of ECs[a], focal disruption of the BBB, microhemorrhages/microinfarcts[a,b,c] | Occlusion of BVs, apoptosis of ECs, focal disruption of the BBB, microhemorrhages/microinfarcts |
| | Ischemic lesions with local loss of neurons, astrocytes, and oligodendrocytes[a,c] | Ischemic lesions with local loss of neurons, astrocytes, and oligodendrocytes |
| **Immune mechanism** | | |
| | Targeting of ECs by CD8+ T cells against unknown EC antigen(s)[a] | Targeting of ECs by MHC-restricted HA-specific CD8+ T cells |
| | No detection of deposition of Ig and complement[a,c] | No detection of deposition of Ig and complement |

Table comparing the neuropathological findings between SuS patients and the EC-HA mouse model
BBB blood–brain barrier, BV blood vessel, CTL cytotoxic T cell, EC endothelial cell, Ig immunoglobulin, ND not detected
[a]This manuscript
[b]Dörr et al., based on MRI findings[7]
[c]Agamanolis et al. and Hardy et al.: based on neuropathological evaluation[9,16]

## Methods

**Patients and controls**. Forty patients with definite and two patients with probable SuS according to the recently defined diagnostic criteria were included in this study[6]. Demographic data of all the patients are summarized in Supplementary Table 1. Twenty of the 42 SuS patients were active at the time point of withdrawal. While 16 of 42 patients were therapy naive, 24 of the 42 patients were on corticosteroids (5 of these patients also received intravenous immunoglobulin (IVIg)), and 2 of 42 patients were on IVIg at the time point of withdrawal. Peripheral blood mononuclear cells (PBMCs) from 262 treatment-naive patients with a stable ($n = 93$) or active ($n = 189$) relapsing–remitting MS fulfilling the revised McDonald criteria[70] and 77 age-and sex-matched healthy donors served as controls. Seventy-six individuals with somatoform disease also served as non-inflammatory controls for CSF analysis. The latter group included patients who were admitted to the hospital because of physical symptoms (i.e., tingling paresthesia) suggesting, e.g., an inflammatory CNS disease, but all clinical, imaging, electrophysiology, and laboratory evaluations were unremarkable. All patients included in this non-inflammatory control group fulfilled the following laboratory criteria defining a non-inflammatory CSF: <5 cells μl$^{-1}$, <500 mg ml$^{-1}$ CSF protein, <2 mM lactate, no disruption of the blood/CSF barrier (defined by the serum/CSF albumin quotient), no intrathecal IgG synthesis (defined by the oligoclonal band pattern), and no intrathecal IgG, IgA, and IgM synthesis (defined by the Reiber' criteria). Six additional MS patients and six patients with no neurological disorders served as controls for the histological staining. All studies and clinical investigations were conducted according to the Declaration of Helsinki and approved by the ethic committee of the University of Münster: registration nos. 2010-262-f-S, 2011-665-f-S, 2014-068-f-S, 2012-407-f-S; and the ethics committee of the Medical University of Vienna: registration nos. 1206/2013 and 1123/2015. All patients provided written formal consent before participating in the study. Lumbar puncture and CNS biopsy were only performed for diagnostic reasons.

As there are no approved treatment options for SuS, treatment with natalizumab, a licensed humanized mAb approved for the therapy of active MS, was used off-label in four patients who did not respond to other immune therapies. All patients planned to receive natalizumab were thoroughly informed about the off-label nature of this approach and their consent was documented in the patients' report by the treating physician.

**Mouse strains**. The double transgenic EC-HA+ mice, expressing HA in CNS microvessel ECs, were generated by crossing the Rosa26$^{tm(HA)1Lib}$ with Slco1c1-CreERT2[39,71] mice, as Slco1c1-CreERT2 mice allow restricted expression in brain ECs. EC-HA+ mice and single transgenic littermate control mice, named EC-HA−, were treated for 5 consecutive days with tamoxifen intraperitoneally (i.p.; 1 mg per mouse per day). The CL4-TCR mice[72] were the source of the HA-specific CD8+ T cells. Both the recipient and the donor mice were bred on the same [BALB/c ×

C57BL/6]F1 background. As recipient mice, for each experiment, we used litters from the first generation of the crossing between Rosa26$^{tm(HA)1Lib}$ BALB/c mice and Slco1c1-CreERT2 C57BL/6 mice. As donor mice, we used CL4-TCR transgenic progeny from the cross of CL4-TCR transgenic mice (backcrossed for >10 times on the BALB/c background) with CD45.1+ C57BL/6 mice. Mice were kept in specific pathogen-free conditions and used in accordance with the European Union guidelines following approval of the local ethics committee (16-U1043 RL/CM-653).

**MRI and CSF routine diagnostics**. MRI was performed on 1.5 or 3 Tesla scanners. Diffusion-weighted imaging with calculation of apparent diffusion coefficient map, axial and coronal T1-weighted spin-echo before and after application of gadolinium, coronal and sagittal fluid-attenuated inversion recovery, axial fast T2-weighted field-echo, and axial turbo T2-weighted spin-echo sequences and diffusion tensor imaging were performed in the patients.

CSF cells were counted using a Fuchs–Rosenthal chamber. Total protein levels, integrity of the blood/CSF barrier, and intrathecal Ig synthesis were analyzed using a nephelometer (Siemens) according to the guidelines by the manufacturer. The oligoclonal band pattern was analyzed by electrophoresis and subsequent silver staining (GE Healthcare).

**Cells and cell culture**. Human blood was collected in EDTA-containing tubes (K2E Vacutainer, BD) and processed within 24 h. PBMCs were isolated by density gradient centrifugation using lymphocyte separation medium (PAA Laboratories).

Immune cells were cultured in complete medium (RPMI 1640 medium supplemented with 10% fetal bovine serum, L-glutamine; all Hyclone) for degranulation assays or in X-Vivo15 (Lonza) for suppression assays. PBMCs for use in TCR Vβ spectratyping and TCR Vβ sequencing were cryopreserved in liquid nitrogen using CTLCABC-cryo freezing media (Immunospot). Frozen PBMCs were thawed by placing vials with frozen cells into a pre-warmed water bath (37 °C) for 8 min. Subsequently, the cell suspension was transferred into a 50-ml conical tube and slowly mixed with pre-warmed RPMI medium supplemented with 10% fetal calf serum (FCS). The cell suspension was centrifuged at $300 \times g$ at room temperature and resuspended in pre-warmed cell culture medium until further processing.

Primary HBMECs (ScienCell) were cultured in speed-coat (Pelobiotech) treated cell-culture flasks in EC medium supplemented with FCS, penicillin/streptomycin, and EC growth supplement (ScienCell).

Mouse Mastocytoma cell line P815 (ATCC) was maintained in Iscove's Modified Dulbecco's Medium/10% FCS.

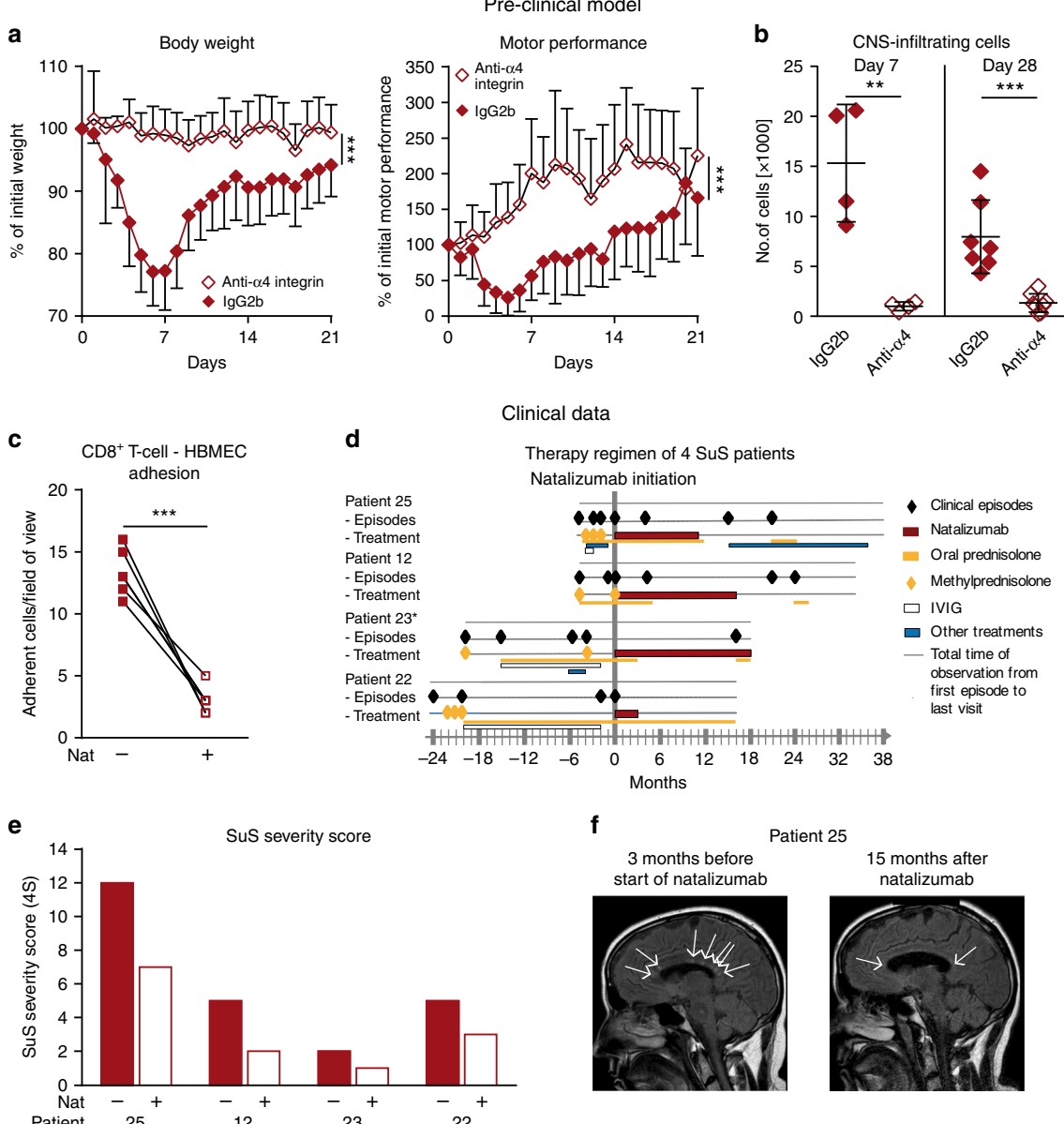

**Fig. 7 Effects of anti-VLA4 mAb in a preclinical and clinical setting. a**, **b** Preclinical data. **a** EC-HA+ mice received HA-specific cytotoxic CD8+ T cells and were treated every 4 days with the anti-mouse α4 integrin mAb (PS/2; open cayenne diamonds) or with an isotype control mAb (IgG2b; closed cayenne diamonds) from day 0 onwards. Clinical signs including weight loss (left) and rotarod performance (right) were assessed daily. The data are from 2 independent experiments involving 10–11/mice per group. **b** Absolute numbers of transferred (Thy1.2+CD45.1+) T cells infiltrating within the CNS of EC-HA+ mice (n = 4–8/group) 7 and 28 days after treatment with a control IgG2b (closed cayenne diamonds) or the anti-α4 integrin mAb (open cayenne diamonds). **c**–**f** Clinical data. **c** Quantification of unstimulated SuS CD8+ T cells (n = 7) adhering to HBMEC monolayer under flow conditions in the absence (closed cayenne squares) or presence (open cayenne squares) of the anti-human α4 integrin mAb natalizumab. Error bars indicate the mean ± SD. **d** Clinical episodes (closed black diamonds) of 4 SuS patients receiving treatments with 300 mg natalizumab monthly (cayenne line), oral prednisolone, tapered from 1 mg/kg body weight (yellow line), pulses of methylprednisolone, 1 g for 5 days (closed yellow diamonds), monthly IVIg 0.4 mg/kg body weight for 5 days (open white line), or other treatments, including cyclophosphamide, plasma exchange, mycophenolate mofetil, and azathioprine (blue line). *Patient 23 is still under treatment with natalizumab, while patients 25, 12, and 22 are not. **e** Bar graph representing the disease score of the four SuS patients before and during treatment with natalizumab. **f** Detection of CNS lesions (arrows) with sagittal FLAIR MRI sequence in patient 25 before (left) and 15 months after the beginning of natalizumab treatment (right). Statistical analysis was performed using unpaired (**b**), paired Student's t test (**c**), or two-way ANOVA with Bonferroni post-test (**a**), respectively. Error bars indicate the mean ± s.d. p values: **p < 0.01; ***p < 0.001. Source data are provided as a Source Data file.

**Purification and isolation of cells**. For TCR spectratyping, sequencing, and physiological flow assays, CD8+ T cells and CD4+ T cells were purified from cryopreserved PBMCs using the CD8+ and CD4+ T cell-negative isolation kits (MACS® magnetic cell separation, Miltenyi Biotech).

For the CDR3-Vβ TCR sequencing of CD8+ T$_{EMRA}$ cells, cryopreserved PBMCs were thawed and CD8+ T$_{EMRA}$ cells were enriched using fluorescence-activated cell

sorting (BD FACSAria™ III). The purity of the sorted CD4+ T cells, CD8+ T cells, and CD8+ T$_{EMRA}$ cells was ≥90%.

For T cell-suppression assays, T$_{reg}$ were isolated by positive selection of CD25+ cells (Miltenyi Biotech) followed by the depletion of CD8+ (2 beads per cell), CD14+ (1 bead per cell), and CD19+ (2 beads per cell) cells using dynabeads (Invitrogen). The remaining cells are enriched for CD4+ T$_{reg}$ cells with >80%

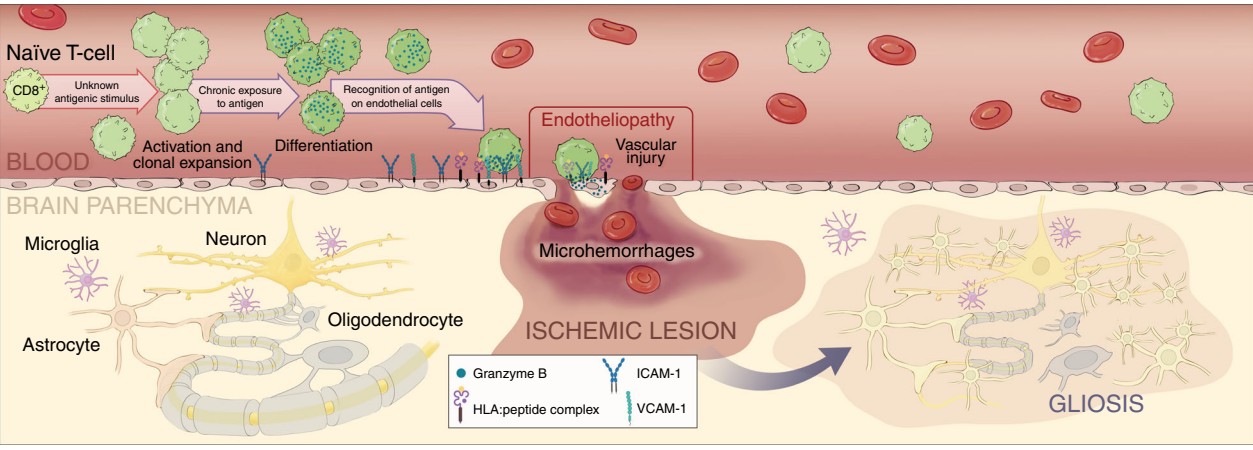

**Fig. 8 Role of CD8+ T cells in the pathophysiology of SuS.** Working model of role of CD8+ T cells in the pathophysiology of SuS. A yet unknown antigenic stimulus drives activation, clonal expansion, and differentiation of CD8+ T cells into GrB- and perforin-expressing CD8+ $T_{EMRA}$ cells. CD8+ T cells accumulate in microvessels of the brain, retina, and inner ear, where they adhere to the endothelium, recognize HLA:peptide complex(es), polarize their cytolytic vesicles toward the endothelial plasma cell membrane, and induce apoptosis of ECs, most likely in a perforin/GrB-dependent manner. Death of ECs and focal disruption of the blood–brain barrier result in microhemorrhages, whereas occlusion of small blood vessels leads to small ischemic lesions with loss of astrocytes, oligodendrocytes, neurons, and axons. Finally, ischemic lesions become gliotic by infiltration of surrounding astrocytes. Illustration©2019-Heike Blum, Department of Neurology with Institute of Translational Neurology, University Hospital Münster.

purity. Effector T ($T_{eff}$) cells were purified from PBMCs by depletion of CD25+ $T_{reg}$ cells using CD25 dynabeads (Invitrogen).

For purification of mouse mononuclear cells, mice were anesthetized with ketamine (100 mg kg$^{-1}$) and xylazine (10 mg kg$^{-1}$) and perfused intracardially with phosphate-buffered saline (PBS). Spleen was collected and dissociated, and red blood cells were lysed. Brains were removed and homogenized, followed by digestion with collagenase D (1 mg ml$^{-1}$), DNase I (10 mg ml$^{-1}$), and TLCK (0.02 mg ml$^{-1}$), and CNS-infiltrating mononuclear cells were then isolated using Percoll density separation, for 30–40 min at 37 °C. After a washing step, mononuclear cells were collected following a Percoll gradient separation. To generate HA-specific CTLs, $20 \times 10^6$ purified naive CD8+ T cells from CD45.1+ CL4-TCR (BALB/c × C57BL/6)F1 mice were stimulated with $100 \times 10^6$ irradiated syngenic splenocytes in Dulbecco's modified Eagle's medium supplemented with 10% FCS containing 1 µg ml$^{-1}$ HA$_{512-520}$ peptide, 1 ng ml$^{-1}$ interleukin (IL)-2, and 20 ng ml$^{-1}$ IL-12. On day 5, cells were collected by Ficoll density separation and $0.3 \times 10^7$ or $3 \times 10^7$ living CTLs were injected IV in recipient mice.

**Flow cytometry and antibodies.** Flow cytometry of whole blood and CSF cells was performed by adding 1 ml VersaLyse buffer (Beckman Coulter) to 100 µl whole blood or CSF cells. Following 10-min incubation at room temperature, 1 ml PBS/0.5% bovine serum albumin (BSA)/2 mM EDTA is added and cells are centrifuged for 4 min at $290 \times g$. Supernatant was discarded and cells were washed once with 3 ml PBS/0.5% BSA/2 mM EDTA. Afterwards cells were stained with the respective fluorochrome-conjugated mAbs (listed in Supplementary Table 5) for 30 min at room temperature. Following one wash step with 1 ml PBS/0.5% BSA/2 mM EDTA, cells were resuspended in 200 µl buffer and 20 µl flow count fluorospheres (Beckman Coulter) were added. Samples were acquired on a Navios flow cytometer (Beckman Coulter). Resulting data were analyzed with Kaluza 1.5a (Beckman Coulter).

For cell-surface staining of PBMCs, cells were stained with fluorochrome-conjugated mAbs (listed in Supplementary Table 5) in PBS/0.5% BSA/2 mM EDTA at 4 °C for 30 min. For the identification of GrB and perforin expression, surface-labeled cells were permeabilized with fixation/permeabilization solutions (BD Cytofix/Cytoperm™) followed by intracellular staining with fluorescence-labeled mAbs for the proteins (Supplementary Table 5). Intra-nuclear staining for Foxp3 was performed using the Foxp3/Transcription Factor Staining Buffer Set (eBioscience) according to the manufacturer's instructions. Naïve/memory T cells were assigned based on CD45RA and CD62L expression for ex vivo analyses. Owing to shedding of CD62L upon thawing, CD27/CD45RA was used to identify naive/memory T cells for thawed cells used for detailed phenotyping and sorting of $T_{EMRA}$ cells. Cell proliferation was tracked by labeling PBMCs with eFluor670 (eBioscience) according to the manufacturer's protocol. Labeled human samples were measured on a Gallios flow cytometer (Beckmann Coulter) and analyzed using Kaluza 1.5a (Beckman Coulter).

For immune cells from EC-HA+ and EC-HA− mice, surface staining was performed with directly labeled antibodies (Supplementary Table 3) in flow cytometric buffer. To assess degranulation and intracellular cytokine production, cells were cultured for 4 h in the presence of 5 µg ml$^{-1}$ allophycocyanin-labeled anti-CD107a antibody and Golgi Stop (BD). Cells were stimulated in vitro with phorbol myristate acetate (1 µg ml$^{-1}$, Sigma), ionomycin (1 µg ml$^{-1}$, Sigma), and

Golgi Plug (BD) for 4 h, followed by intracellular staining for IFN-γ and TNF-α using BD Cytofix/Cytoperm. Mouse data were collected on a LSRII Fortessa flow cytometer (BD) and analyzed with the FlowJo software (Tree Star).

**(Quantitative) histopathology and immunofluorescent staining.** Mice were perfused with PBS followed by 4% paraformaldehyde. Tissues were removed and embedded in paraffin.

Immunohistochemical and immunofluorescence staining was performed on 3–5-µm-thick mouse or human serial sections using the primary antibodies indicated in Supplementary Table 5. Before incubation with primary antibodies, antigen retrieval was performed by heating the sections for 60 min in EDTA (0.05 M) in tris(hydroxymethyl)aminomethane (Tris) buffer (0.01 M, pH 8.5) or citrate buffer (0.01 M, pH 6) in a household food steamer device for all antibodies except for anti-C9 neo and anti-Ig, in which case antigen retrieval was performed by incubating the tissue for 15 min in proteinase (bacterial proteinase Type XXIV, #SLBQ7212V, Sigma Life Science) at 37 °C. For confocal fluorescent double labeling or triple labeling with primary antibodies from different species, primary antibodies were applied simultaneously at 4 °C overnight. After washing with DAKO washing buffer (DakoCytomation, Glostrup, Denmark), Cy2-, Cy3-, or Cy5-conjugated secondary antibodies were applied simultaneously for 1 h at room temperature. For fluorescence triple labeling with two antibodies from the same species, we label the first primary antibody (i.e., mouse-anti-CD31) with a biotinylated secondary antibody and follow-up by tyramide-based amplification of the biotin signal. We then again boil the sections for 30 min in EDTA buffer to completely strip (deactivate) this primary antibody. After this, we incubated with Cy3-conjugated avidin to retrieve the biotin signal, followed by performing a second round of staining with a mixture of primary antibodies from different species (i.e., rabbit-anti-CD8 and mouse-anti-GrB). The staining is finished by Cy2- and Cy5-conjugated secondary antibodies. Fluorescent preparations were embedded and examined using a confocal laser scan microscope (Leica SP5, Leica Mannheim, Germany) equipped with lasers for 504, 488, 543, and 633 nm excitation. Scanning for Cy2 (488 nm), Cy3 (543 nm), and Cy5 (633 nm) was performed sequentially to rule out fluorescence bleed through. Quantification of immunohistochemistry data was performed by counting density of CD3+ cells in various regions of the brain. In each region, the number of T cells in an area of 2 mm$^2$ were counted by using a morphometric grid. Optical density of Ig leakage of total brain sections or individual cortical or hippocampal regions was analyzed using Image J. Quantification of immune cells of human immunohistochemistry data was performed by counting cells using a morphometric grid. Owing to the different sizes of the specimens, areas analyzed ranged from 5 to 48 mm$^2$ (on average 14 mm$^2$). Quantification of apoptosis of ECs was performed by analysis of nuclear condensation of CD34+ ECs. For this, SuS, MS lesions, and control sections were investigated at ×400 magnification, counting a minimum of 50 cells (in the smallest specimen) to a maximum of 250 cells. The same method was used for analysis of apoptosis in the cerebelli of mice. Here on average 120 ECs were counted. Attachment of CD8+ CTLs to blood vessels was performed on SuS and MS sections double labeled with CD8 and CD34. Parenchymal CD8+ CTLs and CD8+ CTLs attached to the luminal or perivascular side of CD34 stained ECs were counted and the percentage of attachment was calculated. On average, 99 CTLs in

Susac and 194 cells in MS were counted. Fluorescent preparations were examined using a confocal laser scanning microscope (Leica SP5).

**RNA isolation and reverse transcriptase-PCR**. For TCR profiling, RNA was isolated from human CD8$^+$ T cells, CD4$^+$ T cells, and CD8$^+$ T$_{EMRA}$ cells using the RNAeasy$^{TM}$ Kit (Qiagen) according to the manufacturer's instructions. cDNA was prepared from 0.5 to 1 μg isolated RNA by reverse transcription using reagents including reverse transcriptase, dNTPs, and random hexamers (Life Technologies). To assess HA gene expression in mice, total RNA was extracted from mouse tissue with the RNeasy Mini-kit or RNeasy Micro-kit plus (Qiagen), followed by reverse transcription using RevertAidH Minus Revert Transcriptase (ThermoFisher).

The cDNA prepared from mouse tissue was used as a template for quantitative PCR using SYBR Green I master mix (Roche) and assessed in a thermocycler (LightCycler 480; Roche). mRNA expression was normalized to that of HPRT mRNA. Specific primers were used to amplify HA (forward, 5′-AAACTCTTCG CGGTCTTTTCCA-3′; reverse, 5′-GATAAGGTAGCTTGGGCTGC-3′), and HPRT (forward, 5′TGGTTAAGCAGTACAGCCCCAA-3′; reverse, 5′-AGGTCCTTTTCA CCAGCAAGCT).

**TCR spectratyping**. Spectratyping of the CDR3 sequences of TCR-Vβ-chains of CD4$^+$ and CD8$^+$ T cell subsets was performed by multiplex PCR using 24 Vβ forward primers and Cβ reverse primers "SpTy-b-out" and Cβ-R (provided by Metabion, Martinsried, Germany)[73,74]. Data analysis of the spectratypes was performed using the GeneMarker software (SoftGenetics, State College, PA, USA). Complexity score was calculated based on the number of peaks across all 24 spectratypes per individual. A lower complexity score is indicative of a more perturbed repertoire. Fragment length distribution classification was determined based on the number of observed peaks and the area under the curve for each peak in a spectratype. Gaussian distributed spectratypes were considered "normal"; "shifted" spectratypes displayed all peaks with shifted maxima, "skewed" spectratypes show reduced peak numbers and non-Gaussian distributions, and spectratypes with only 1–2 peaks that were twice the height of comparable peaks of a normally distributed spectratype were considered "oligoclonal."

**High-throughput TCR sequencing**. TCRs of CD8$^+$ and CD4$^+$ T cells isolated from the blood of HD, SuS, and MS patients were profiled with high-throughput TCR sequencing. The ImmunoSEQ assay (Adaptive Biotechnologies), a multiplex PCR system, was used to amplify 54 expressed Vβ and 13 Jβ segments. The resulting amplicons span and sufficiently identify the VDJ region of each unique CDR3β[22]. PCR bias was corrected using a baseline platform developed from a suite of synthetic templates to optimize primer concentrations and for computational corrections. The amplicons were deep sequenced using the Illumina HiSeq platform. The raw reads were annotated into V, D, and J gene definitions according to the IMGT (ImmunoGeneTics) database (http://www.imgt.org). The TCR reads were pre-processed and data were analyzed with the ImmunoSEQ Analyzer software. Productive sequences excluding those that were out of frame or have a stop codon were considered for all analyses. Perturbations in the repertoire were computed as clonality. Clonality is based on Shannon's entropy, wherein clonality is equivalent to 1 − normalized entropy (normalized entropy = entropy/log2 (productive unique)). Clonality ranges from 0 to 1 where a value closer to 1 represents a repertoire dominated by select clones. Identification of private/public clones was performed as previously published[24]. SuS-specific clones were defined as clones absent in 27 control HD ($n = 12$) and MS cohorts ($n = 15$), 1052 published viral clones including 956 CMV clones, and other published sequences curated in the ImmuneACCESS database. Private clones were defined as those clones exclusively found in one individual while public clones are non-exclusive to the individual.

**Degranulation assay**. To assess the release of cytotoxic granules of CD8$^+$ T cell in response to polyclonal stimulation by anti-CD3-loaded on P815 cells[75], PBMCs were mixed in a 1:1 ratio with P815 target cells and incubated in the presence of 10 μl ml$^{-1}$ Golgi plug (BD Bioscience), 0.0625 μg ml$^{-1}$ anti-CD3 (Biolegend), and 0.25 μg ml$^{-1}$ CD107a-AF488 antibody (Biolegend) for 3 h at 37 °C/5% CO$_2$. Following incubation, cells were centrifuged at 300 × g for 10 min at room temperature and stained for lineage-defining surface markers for 30 min at 4 °C. Following washing, samples were acquired by flow cytometry. Degranulation of CD8$^+$ T cells was assessed as the percentage of CD107a$^+$-expressing CD8$^+$ T cells.

**T cell-suppression assay**. For T cell-suppression assays, 5 μM eFluor-670 (eBioscience) labeled T$_{eff}$ cells were cultured either alone or co-cultured with titrated numbers of T$_{regs}$ in 1:1 to 1:0.08 ratio. Therefore, T$_{eff}$ were isolated from PBMCs by depletion of CD25-expressing cells using CD25 dynabeads (Invitrogen) as described above. T$_{eff}$ cells were labeled with eFluor-670 according to the manufacturer's instructions. T$_{eff}$ cells were mixed with T$_{reg}$ isolated from independent donors and were polyclonally stimulated with 0.5 μg ml$^{-1}$ aCD3 (clone OKT3, Biolegend) in X-Vivo15 medium (Lonza) for 4 days at 37 °C/5% CO$_2$. Following incubation, cells were stained with lineage-defining fluorochrome-conjugated antibodies and acquired by flow cytometry. Suppression of T$_{eff}$ proliferation was

evaluated by tracking the dilution of eFluor-670 dye on CD4$^+$ and CD8$^+$ T$_{eff}$ cells in cultures in comparison to cultures with and without T$_{reg}$.

**Physiological flow assay**. Physiological flow assays were performed in flow chambers (0.4 μm slides; IBIDI) with monolayers of HBMECs (Pelobiotech). HBMECs were treated with TNF-α (500 U ml$^{-1}$; R&D Systems) for 18 h. CD8$^+$ T cells (3.5 × 10$^5$ cells per slide) isolated from thawed PBMCs of healthy donors or SuS patients were perfused at a constant shear stress of 0.25 dyn cm$^{-2}$ for 5 min. If indicated, T cells were incubated with natalizumab (Tysabri, Biogen; 10 μg ml$^{-1}$) for 10 min. Videos (×20 magnification) were recorded using a BZ-9000 BioRevo microscope (Keyence) and BZ II viewer software (Keyence). Adherent cells per field of view were determined using ImageJ (NIH).

**Transfer of HA-specific CTLs**. On day 5 of differentiation of HA-specific CTLs, cells were collected by Ficoll density separation and 0.3 × 10$^7$ and 3 × 10$^7$ CTLs were adoptively transferred IV to EC-HA$^+$ mice and EC-HA$^-$ littermate controls. Mice were assessed daily for weight and neurological signs. The motor skill and performance were assessed using a Rotarod device (Bioseb), with acceleration from 4 to 40 rpm over a 600-s period. First, the animals were trained before treatment to reach steady performance. The 1-week training consisted of 2–3 trials of 10 min each with rest of 5 min between the trials. During the investigation period (days 0–28), the amount of time spent on the drum was measured daily for each mouse. 100% represents the day-0 value.

**Antibody treatment of mice**. The anti-α4 integrin antibody (clone PS/2; BioXCell, West Lebanon, NH, USA) or a control rat IgG2b was injected into EC-HA$^+$ mice (250 μg per mouse i.p.) at the indicated time points.

**Natalizumab treatment of patients**. The humanized anti-α4 integrin antibody (natalizumab (Tysabri™), Biogen, Boston USA), which blocks VLA-4 from interacting with VCAM-1, was infused monthly at a fixed 300 mg dose of natalizumab every 28 days.

**Statistical analysis**. Statistical analysis was performed using the GraphPad Prism 5.0 software. Data in figures are represented as means ± standard deviation (s.d.). D'Agostino-Pearson omnibus normality test was performed to test for Gaussian distribution. Statistical significance for normally distributed data was determined using unpaired, two-tailed Student's $t$ tests, while Mann–Whitney test was used for non-parametric data to compare means between two independent groups. For comparison of more than two groups, one-way analysis of variance with Bonferroni post-test (parametric) or Kruskal–Wallis with Dunn's post-test (non-parametric) was applied. Paired Student's $t$ test or Wilcoxon matched test was used for different treatments within the same patient group. Correlation analysis was performed using linear regression. $p$ Values of <0.05 were considered significant and indicated in the corresponding figures (*$p < 0.05$; **$p < 0.01$; ***$p < 0.001$; ****$p < 0.0001$).

**Reporting summary**. Further information on research design is available in the Nature Research Reporting Summary linked to this article.

## Data availability
There is no restriction in the availability of materials described in the study. Source data are provided as a Source Data file. TCR sequences have been deposited to the BioProject database under the accession number PRJNA579190.

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

## Acknowledgements

We thank all patients included in this study for their participation. We also thank Daniela Roostermann, Emilie Mauré, Gabriele Berens, Kirsten Weiß, Julia Sundermeier, Lena Schünemann, Aline Kiese, Arne Seeger, Alina Teich, Tobias Schilling, Ulrike Köck, Angela Kury, Marianne Leisser, and Christiane Schulze-Weppel for their excellent technical assistance. The authors also thank Thomas Korn for discussion and helpful suggestions regarding this manuscript as well as Heike Blum for designing Supplementary Fig. 10. This work was supported by the European Susac Consortium (EuSaC); the German Research Foundation grant (DFG) GR3946_3/1 "Susac syndrome (SuS) as a paradigm of a CD8+ T cell mediated endotheliopathy" (to C.C.G. and I.K.); the Interdisciplinary Center for Clinical Studies (IZKF) grant Kl3/010/19 "Antigen-specific CD8 T cell responses in Neuromyelitis Optica and Susac syndrome" (to L.K. and C.C.G.); the IMF grant KL 111421 "The pathophysiological role and clinical impact of the immune system in Susac syndrome (SuS)" (to I.K. and C.C.G.); the Collaborative Research Centre CRC TR128 "Initiating/Effector versus Regulatory Mechanisms in Multiple Sclerosis—Progress towards Tackling the Disease" (projects A05 to K.D., A09 to H.W. and C.C.G., B01 to N.S., and Z02 to H.W. and T.K.); SFB1009 Breaking Barriers (project A03 to H.W. and L.K.); the Federal Ministry of Education and Research funded Disease Related Competence Network for Multiple Sclerosis (KKNMS, project FKZ01FI1603A to H.W., L.K., and C.C.G.); the intramural Cell in Motion (CiM) cluster of excellence bridging fund to U.B.; the Oppenheim Research Grant MS to A.S.-M. and C.C.G.; the German Research Council through the Munich Cluster for Sastems Neurology (SyNergy EXC 2145, project 390857198 to K.D. The design and study of the mouse model was supported by Inserm, CNRS, Toulouse University and by grants from the French MS society (ARSEP), the Foundation pour la Recherche Médicale (FRM), the French Research Agency (ANR T cell Mig), ERA-NET NEURON (Meltra-BBB), the Institut Universitaire de France, and the German Research Foundation (DFG, SCHW 416/5–2 to M.S.). J.B.'s work is supported by the Austrian Science Fund (FWF: P26936-B27). R.H.'s work was supported by the Austrian Science Fund (FWF: I3334-B27).

## Author contributions

C.C.G. originated and designed all human parts of the study, supervised CSF analysis, designed, generated and supervised experiments using human biomaterial (blood and CSF), analyzed and interpreted data, discussed results and implications at all stages of the study, designed the figures, and wrote the manuscript. C.M. generated the mouse model, performed the related in vivo experiments, analyzed the resulting data, drafted the figures, figure legends, and critically revised the manuscript. U.B. performed TCR spectratyping, high-throughput TCR sequencing and established and performed human phenotyping and functional assays, analyzed and interpreted the resulting data, drafted the figures, figure legends, and helped writing parts of the manuscript. L.Y. performed mouse in vivo experiments, RT-qPCR, flow cytometry assays and the PS/2 treatment in the mouse model, analyzed and interpreted the resulting data, drafted the figures, figure legends, and critically revised the manuscript. I.K. recruited patients, took patient care, designed and performed the natalizumab study, analyzed and interpreted clinical data, and critically revised the manuscript. J.B. performed and interpreted the quantitative analysis of T cell numbers, interactions of T cells with ECs and apoptosis of ECs as well as immunohistochemical and confocal fluorescence microscopy data of experimental animals and human brain biopsies. He also critically revised the manuscript. A.T. analyzed immunoglobulin leakage in experimental animals. A.S.-M. performed experiments, analyzed data, and drafted the figures. S.H. performed physiological flow assays and

analyzed data. T.S.-H. discussed TCR sequencing data. H.P. performed experiments and analyzed data. T.K. analyzed and interpreted immunohistology data. M.S. provided expertise in generating the mouse model and discussed results and implications of these data. W.B. analyzed and interpreted immunohistology data. M.P., D.-A.L., D.L., M.R., J.D., B.W., and M.K. recruited SuS patients, took patient care, and interpreted clinical data. J.P., M.B., T.A.H., and S.W.R. recruited SuS patients, took patient care, interpreted clinical data, and critically revised the manuscript. M.E.B. performed and interpreted immunohistochemical data on human SuS brain biopsies, and critically revised the manuscript. H.L. analyzed and interpreted neuropathological and immunohistology data. R.H. performed the neuropathological and immunohistochemical analysis of human SuS syndrome brain. E.B. discussed and supported interpreting TCR sequencing data. K.D. provided his expertise in TCR spectratyping and sequencing and discussed results and implications of these data in all stages of the study. N.S. designed, supervised, and interpreted physiological flow assays. He also discussed spectratyping and sequencing data. L.K. discussed data and critically revised the manuscript. S.G.M. designed and supervised the natalizumab trial, interpreted clinical data, discussed results and implications at all stages of the study, and critically revised the manuscript. G.M.-B. designed and supervised the murine part of the study, discussed results and implications during the course of the study, and critically revised the manuscript. H.W. originated the project, designed all human parts of the project, and supervised the experimental and clinical parts of the study using human biomaterial (blood, CSF) and involving patients, respectively, discussed results and implications at all stages of the study, and critically revised the manuscript. R.L. originated, designed, and supervised the murine part of the study; discussed results and implications during the course of the study; and critically revised the manuscript. All authors read and approved the final version of the manuscript.

## Competing interests

C.C.G. received speaker honoraria and travel expenses for attending meetings from Biogen, Euroimmun, Genzyme, MyLan, Novartis Pharma GmbH, and Bayer Health Care, none related to this study. Her work is funded by Biogen, Novartis, the German Ministry for Education and Research (BMBF; 01Gl1603A), the German Research Foundation (DFG, GR3946/3–1, SFB Transregio 128 A09), the European Union (Horizon2020, ReSToRE), the Interdisciplinary Center for Clinical Studies (IZKF), and the IMF. I.K. received travel expenses for attending meetings from Pfizer and CSL Behring. I.K. received speaker honoraria from Daiichi Sankyo. J.B.'s work is funded by the Austrian Science Fund (FWF: P26936-B27). A.S.-M. receives research support from Novartis. T.K. received speaker honoraria from Novartis and Excemed. Her research is supported by the DFG (SFB/Transregio 128 B07), European Leukodystrophy Association (2017-018C4A), National MS Society (RG-1801-30020), and Progressive MS Alliance (PA-1604-08492, VRAVEinMS). W.B. has received honoraria for lectures by Bayer Vital, Biogen, Merck Serono, Teva Pharma, Genzyme, Sanofi-Aventis, Novartis, Excemed, and Medday. He is a member of scientific advisory boards for Teva Pharma, Biogen, Novartis, and Genzyme. W.B. receives research support from Teva Pharma, Biogen, Genzyme, and Novartis. W.B. is supported by the Deutsche Forschungsgemeinschaft, the German Ministry for Education and Research, and by the Klaus Tschira Foundation. M.P. received speaker honoraria from Roche and Genzyme and travel/accommodation/ meeting expenses from Novartis, Biogen Idec, Genzyme, and MERCK Serono. D.-A.L. received speaker honoraria from Merck, Novartis, Roche, Teva, and Genzyme and non-personal grants from Biogen, Genzyme, Novartis, MedDay, Merck, and Roche not related to this study. J.P. is a member of scientific advisory boards from Genzyme and Novartis. He received honoraria from Biogen, Sanofi/Genzyme, and Merck Serono. M.B. received institutional support for research, speaking and/or participation in advisory boards for Biogen, Merck, Novartis, Roche, and Sanofi Genzyme. He is a consulting neurologist for RxMx/Medical Safety Systems and a research director for Sydney Neuroimaging Analysis Centre. T.A.H. has received honoraria or travel sponsorship from Bayer-Schering, Novartis, Biogen Idec, Merck-Serono, Roche, Teva, Alexion, and Sanofi-Genzyme. S.W.R. has received honoraria, travel sponsorship, research, and/or departmental support from MGANSW, MGAQLD, MAA, Lambert Initiative, Beeren foundation, anonymous donors, and from pharmaceutical/biological companies: Baxter, Bayer Schering, Biogen Idec, CSL, Genzyme, Grifols, Octapharma, Merck, Novartis, Roche, Sanofi Aventis Genzyme, Servier, and TEVA. Relevant to this study, Biogen is the manufacturer of natalizumab. S.W.R. is a shareholder of Medical Safety Systems trading as RxMx (grant and contracts with Genzyme >$25,000 AUD, contracts with Novartis, Roche, Janssen). M.R. received speaker honoraria from Novartis, Bayer Vital GmbH, and Ipsen and travel reimbursement from Bayer Schering, Biogen Idec, Merz, Genzyme, Teva, and Merck, none related to this study. J.D. received research support by Bayer and Novartis, travel support by Bayer, Novartis, Biogen, and Merck Serono, and honoraria for lectures and advisory by Bayer, Novartis, Biogen, Merck Serono, Roche, and Sanofi Genzyme. B.W. received research support from German Ministry of Education and Research, Dietmar Hopp Foundation, Klaus Tschira Foundation, Sanofi Genzyme, Merck Serono, and Novartis and speaker honoraria and/or travel support from Bayer Healthcare, Biogen, Merck Serono, Novartis, Sanofi Genzyme, and TEVA outside the submitted work. M.K. received travel support and honoraria from Bayer Schering, Biogen Idec, Chugai Pharma, Merck Serono, Novartis Pharma, Teva Pharma, and Shire Deutschland. H.L. received honoraria for lectures and consultation related to multiple sclerosis from Novartis, Roche, Sanofi Aventis, Biogen, and MEDDAY. R.H. received speaker honoraria from Euroimmun and research support from the Jubiläumsfonds der Östereichischen Nationalbank (project 16919) and the Austrian Science Fund (FWF: I3334-B27). N.S.

received travel support from Novartis and Sanofi-Genzyme. L.K. received compensation for serving on scientific advisory boards for Genzyme, Merck, Novartis, and Roche; speaker honoraria and travel support from Biogen, Genzyme, Merck, Novartis, and Roche; and research support from Biogen, Genzyme, Merck, Novartis, and Roche, the German Ministry for Education and Research (BMBF), Deutsche Forschungsgesellschaft (DFG), and the Interdisciplinary Center for Clinical Studies (IZKF) Münster. S.G.M. receives honoraria for lecturing, and travel expenses for attending meetings from Almirall, Amicus Therapeutics Germany, Bayer Health Care, Biogen, Celgene, Diamed, Genzyme, MedDay Pharmaceuticals, Merck Serono, Novartis, Novo Nordisk, ONO Pharma, Roche, Sanofi-Aventis, Chugai Pharma, QuintilesIMS, and Teva. His research is funded by the German Ministry for Education and Research (BMBF), Deutsche Forschungsgesellschaft (DFG), Else Kröner Fresenius Foundation, German Academic Exchange Service, Hertie Foundation, Interdisciplinary Center for Clinical Studies (IZKF) Muenster, German Foundation Neurology and Almirall, Amicus Therapeutics Germany, Biogen, Diamed, Fresenius Medical Care, Genzyme, Merck Serono, Novartis, ONO Pharma, Roche, and Teva. G.M.-B. received speaker honoraria and travel support for attending meetings from Abbvie, Genzyme, Gilead, and Pfizer. H.W. received honoraria for scientific advisory boards/steering committees from Biogen, Evgen, MedDay Pharmaceuticals, Merck Serono, Novartis, Roche Pharma AG, and Sanofi-Genzyme. He received speaker honoraria and travel support for attending meetings from Alexion, Biogen, Cognomed, F. Hoffmann-La Roche Ltd., Gemeinnützige Hertie-Stiftung, Merck Serono, Novartis, Roche Pharma AG, Sanofi-Genzyme, TEVA, and WebMD Globa. H. W. received compensation as a consultant from Abbvie, Actelion, Biogen, IGES, Novartis, Roche, Sanofi-Genzyme, and the Swiss Multiple Sclerosis Society. He also received research support from the German Ministry for Education and Research (BMBF), Deutsche Forschungsgesellschaft (DFG), Else Kröner Fresenius Foundation, Fresenius Foundation, Hertie Foundation, NRW Ministry of Education and Research, Interdisciplinary Center for Clinical Studies (IZKF) Muenster and RE Children's Foundation, Biogen GmbH, GlaxoSmithKline GmbH, Roche Pharma AG, and Sanofi-Genzyme. R.L. received grant support from Pierre Fabre, GlaxoSmithKline, and Diaccurate. He received speaker or scientific board honoraria from Biogen, Servier, Novartis, and Sanofi-Genzyme. R.L. is currently receiving grants from GlaxoSmithKline, Cancer Research Institute, French Cancer research foundation (ARC), Rare Diseases Foundation, and National Institute of Cancer (INCa). C.M., U.B., L.Y., A.T., S.H., T.S.-H., H.P., M.S., D.L., M.E.B., E.B., and K.D. have no financial disclosures.

## Additional information

Catharina C. Gross [1,26]*, Céline Meyer[2,26], Urvashi Bhatia [1,26], Lidia Yshii[2,26], Ilka Kleffner [1,3,26], Jan Bauer [4,26], Anna R. Tröscher [4], Andreas Schulte-Mecklenbeck[1], Sebastian Herich [1], Tilman Schneider-Hohendorf [1], Henrike Plate [1], Tanja Kuhlmann [5], Markus Schwaninger[6], Wolfgang Brück [7], Marc Pawlitzki [1], David-Axel Laplaud [8,9], Delphine Loussouarn[10], John Parratt[11,12], Michael Barnett[13], Michael E. Buckland [13,14], Todd A. Hardy[13,15], Stephen W. Reddel[13,15], Marius Ringelstein [16,17], Jan Dörr[18], Brigitte Wildemann[19], Markus Kraemer [16,20], Hans Lassmann [4], Romana Höftberger[21], Eduardo Beltrán [22], Klaus Dornmair [22,23], Nicholas Schwab [1], Luisa Klotz [1], Sven G. Meuth [1,24,27], Guillaume Martin-Blondel[2,25,27], Heinz Wiendl[1,12,24,27]* & Roland Liblau[2,27]*

[1]Department of Neurology with Institute of Translational Neurology, University Hospital Münster, University of Münster, Albert-Schweitzer-Campus 1, 48149 Münster, Germany. [2]Centre de Physiopathologie Toulouse-Purpan (CPTP), Université de Toulouse, CNRS, Inserm, UPS, CHU Purpan – BP 3028 – 31024, Toulouse Cedex 3, Toulouse, France. [3]Department of Neurology, University Hospital Knappschaftskrankenhaus Bochum, Ruhr University Bochum, In der Schornau 23-25, 44892 Bochum, Germany. [4]Department of Neuroimmunology, Center for Brain Research, Medical University of Vienna, Spitalgasse 4, 1090 Vienna, Austria. [5]Institute of Neuropathology, University Hospital Münster, University of Münster, Pottkamp 2, 48149 Münster, Germany. [6]Institute of Experimental and Clinical Pharmacology and Toxicology, University of Lübeck, Ratzeburger Allee 160, 23562 Lübeck, Germany. [7]Institute of Neuropathology, University Medical Center Göttingen, Robert-Koch-Straße 40, 37099 Göttingen, Germany. [8]UMR 1064, INSERM, Centre de Recherche en Transplantation et Immunologie, Université de Nantes, CHU Nantes - Hôtel Dieu Bd Jean Monnet, 44093 Nantes Cedex 01, France. [9]Service Neurologie, CHU Nantes, Nantes, France. [10]Service d'Anatomo-Pathologie, CHU Nantes, Hôtel-Dieu, rez-de-jardin, 44093 Nantes Cedex 1, France. [11]Department of Neurology, Royal North Shore Hospital, Sydney, Australia. [12]Australia Northern Clinical School, University of Sydney, Reserve Road, St Leonards, Sydney, NSW 2065, Australia. [13]Brain and Mind Centre, Medical Faculty, University of Sydney, Mallett Street, Camperdown, Sydney, NSW 2050, Australia. [14]Department of Neuropathology, Royal Prince Alfred Hospital, 94, Mallett Street, Camperdown, Sydney, NSW 2050, Australia. [15]Department of Neurology, Concord Hospital, University of Sydney, Sydney, NSW 2139, Australia. [16]Department of Neurology, Medical Faculty, Heinrich Heine University, Moorenstraße 5, 40225 Düsseldorf, Germany. [17]Department of Neurology, Center of Neurology und Neuropsychiatry, LVR-Klinikum, Heinrich Heine University Düsseldorf, Bergische Landstraße 2, 40629 Düsseldorf, Germany. [18]Max Delbrueck Center for Molecular Medicine and Charité – Universitätsmedizin Berlin, corporate member of Freie Universität Berlin, Humboldt-Universität zu Berlin, and Berlin Institute of Health, NeuroCure, Experimental and Clinical Research Center, Charitéplatz 1, 10117 Berlin, Germany. [19]Molecular Neuroimmunology Group, Department of Neurology, University of Heidelberg, Im Neuenheimer Feld 400, 69120 Heidelberg, Germany. [20]Department of Neurology, Alfried Krupp Hospital, Alfried-Krupp-Strasse 21, 45130 Essen, Germany.

[21]Institute of Neurology, Medical University of Vienna, Währinger Gürtel 18-20, 1090 Vienna, Austria. [22]Institute of Clinical Neuroimmunology, Biomedical Center and Hospital of the Ludwig-Maximilians-University Munich, Großhaderner Straße 9, Martinsried, 82152 Munich, Germany. [23]Munich Cluster for Systems Neurology (SyNergy), Munich, Germany. [24]Cells in Motion (CiM), Münster, Germany. [25]Department of Infectious and Tropical Diseases, Toulouse University Hospital, Toulouse, France. [26]These authors contributed equally: Catharina C. Gross, Céline Meyer, Urvashi Bhatia, Lidia Yshii, Ilka Kleffner, Jan Bauer [27]These authors jointly supervised this work: Sven G. Meuth, Guillaume Martin-Blondel, Heinz Wiendl, Roland Liblau *email: Catharina.Gross@ukmuenster.de; Heinz.Wiendl@ukmuenster.de; Roland.Liblau@inserm.fr

