## [Peer Review File · Nature Communications]

Editorial Note: This manuscript has been previously reviewed at another journal that is not operating a transparent peer review scheme. This document only contains reviewer comments and rebuttal letters for versions considered at *Nature Communications* .

REVIEWERS' COMMENTS:

Reviewer #1 (Remarks to the Author):

The revisions in this ms since my original review have adequately addressed my major concern, namely that the analysis of patient pathological specimens was inadequate, failing even to note the number of specimens examined. The revised ms has clearly expanded this aspect of the work, improving the paper by doing so. My sole current reservation is that I feel the authors are over-interpreting the mouse model. For example, human endothelial cells differ from mouse ECs in their capacity to express class II MHC molecules and to activated effector memory CD4+ T cells. Consequently, mouse models can fail to reveal a role for CD4+ T cells. I would suggest the authors be a bit more circumspect in the discussion.

Reviewer #2 (Remarks to the Author):

The authors have addressed all of my comments and the paper is now ready for acceptance.

Reviewer #3 (Remarks to the Author):

The authors have done an excellent job addressing the reviewers' concerns. I have one additional comment regarding the use of Slco1c1-CreERT2 x ROSA26-Stop-HA mice. In response to Reviewer#3, the authors did not provide data showing the CNS expression pattern of the Slco1c1-CreERT2 mice in their hands. This is important because the authors assume that the phenotype observed in Slco1c1-CreERT2 x ROSA26-Stop-HA mice is linked exclusively to the expression of HA in cerebrovascular endothelial cells. While the promoter does not appear to be active in most peripheral tissues tested, it is active in the brain, retina, and inner ear. Within the brain, expression has been reported in hippocampal neurons, astrocytes, and choroid plexus epithelial cells, making it difficult to interpret the pathology and symptom data in figures 4 and 5. In the revised manuscript, the authors add the retina and inner ear to the list of tissues that express Slco1c1-CreERT2, but they don't show the expression pattern. If the authors are not planning to show the cellular expression patterns in the three positive compartments, then it is important to state in the manuscript that the phenotypes cannot be linked exclusively to T cell engagement of vascular endothelial cells.

Nature Communications Manuscript NCOMMS-19-28933-A

Point-by-Point Reply - Revision

We would like to thank all reviewers for their helpful comments. The resulting changes in the manuscript are highlighted in red.

Reviewer #1:

The revisions in this ms since my original review have adequately addressed my major concern, namely that the analysis of patient pathological specimens was inadequate, failing even to note the number of specimens examined. The revised ms has clearly expanded this aspect of the work, improving the paper by doing so. My sole current reservation is that I feel the authors are over-interpreting the mouse model. For example, human endothelial cells differ from mouse ECs in their capacity to express class II MHC molecules and to activated effector memory CD4⁺ T cells. Consequently, mouse models can fail to reveal a role for CD4⁺ T cells. I would suggest the authors be a bit more circumspect in the discussion.

We have now made text changes (in red in the discussion) to be a bit more circumspect in the discussion and acknowledge that the role of CD4⁺ T cells still has to be fully investigated.

Reviewer #2:

The authors have addressed all of my comments and the paper is now ready for acceptance.

Reviewer #3:

The authors have done an excellent job addressing the reviewers' concerns. I have one additional comment regarding the use of Slco1c1-CreERT2 x ROSA26-Stop-HA mice. In response to Reviewer#3, the authors did not provide data showing the CNS expression pattern of the Slco1c1-CreERT2 mice in their hands. This is important because the authors assume that the phenotype observed in Slco1c1-CreERT2 x ROSA26-Stop-HA mice is linked exclusively to the expression of HA in cerebrovascular endothelial cells. While the promoter does not appear to be active in most peripheral tissues tested, it is active in the brain, retina, and inner ear. Within

the brain, expression has been reported in hippocampal neurons, astrocytes, and choroid plexus epithelial cells, making it difficult to interpret the pathology and symptom data in figures 4 and 5. In the revised manuscript, the authors add the retina and inner ear to the list of tissues that express Slco1c1-CreERT2, but they don't show the expression pattern. If the authors are not planning to show the cellular expression patterns in the three positive compartments, then it is important to state in the manuscript that the phenotypes cannot be linked exclusively to T cell engagement of vascular endothelial cells.

We have now made text changes (in red in the discussion) to take into consideration the recommendation of the Reviewer to indicate that the phenotypes may not be exclusively linked to T cell engagement of BBB-endothelial cells.